# Short-term hypercaloric carbohydrate loading increases surgical stress resilience by inducing FGF21

Thomas Agius [1,2,3,12], Raffaella Emsley[1,12], Arnaud Lyon [1], Michael R. MacArthur [4], Kevin Kieswo ro[1], Anna Faivre[5,6], Louis Stavart [7], Martine Lambelet [1], David Legouis [5,8], Sophie de Seigneux [5,6], Déla Golshayan [7], Francois Lazeyras[9,10], Heidi Yeh[2,3], James F. Markmann [2,3], Korkut Uygun[2,3], Alejandro Ocampo [11], Sarah J. Mitchell[4], Florent Allagnat [1,11], Sébastien Déglise[1] & Alban Longchamp [1,2,3] ✉

Dietary restriction promotes resistance to surgical stress in multiple organisms. Counterintuitively, current medical protocols recommend short-term carbohydrate-rich drinks (carbohydrate loading) prior to surgery, part of a multimodal perioperative care pathway designed to enhance surgical recovery. Despite widespread clinical use, preclinical and mechanistic studies on carbohydrate loading in surgical contexts are lacking. Here we demonstrate in *ad libitum*-fed mice that liquid carbohydrate loading for one week drives reductions in solid food intake, while nearly doubling total caloric intake. Similarly, in humans, simple carbohydrate intake is inversely correlated with dietary protein intake. Carbohydrate loading-induced protein dilution increases expression of hepatic fibroblast growth factor 21 (FGF21) independent of caloric intake, resulting in protection in two models of surgical stress: renal and hepatic ischemia-reperfusion injury. The protection is consistent across male, female, and aged mice. In vivo, amino acid add-back or genetic FGF21 deletion blocks carbohydrate loading-mediated protection from ischemia-reperfusion injury. Finally, carbohydrate loading induction of FGF21 is associated with the induction of the canonical integrated stress response (ATF3/4, NF-kB), and oxidative metabolism (PPARγ). Together, these data support carbohydrate loading drinks prior to surgery and reveal an essential role of protein dilution via FGF21.

Worldwide, 310 million surgeries are performed annually, with approximately 40 to 50 million taking place in the USA and 20 million in Europe. These surgeries carry a postoperative mortality rate of 1–4%, a morbidity rate of up to 15%, and a readmission rate within 30 days ranging from 5–15%[1]. With an estimated global annual mortality of 8 million, the mortality associated with major surgery is comparable to that of cardiovascular disease (17.9 million), stroke (5 million), and cancer (10 million)[1]. Ischemia-reperfusion injury (IRI) is an important component of surgical stress, involving the occlusion of blood flow to an organ or tissue for a certain period of time (ischemia) and subsequent restoration of blood supply (reperfusion)[2]. IRI represents a major clinical concern in controlled elective surgery, which requires temporary restriction of blood flow (e.g. solid organ transplantation, vascular surgery) and uncontrolled settings (stroke, heart attack, limb

---

trauma). During the ischemic period, cells are deprived of oxygen and nutrients, resulting in ATP depletion, loss of membrane potential, and the accumulation of toxic byproducts. Subsequent reperfusion introduces additional damage via extensive ROS generation by the reverse electron transport at the mitochondria[3]. Together, there is a significant need for interventions that can reduce IRI and enhance resilience to surgical stress.

Currently, perioperative surgical care in humans includes enhanced-recovery protocols (ERAS®), which encourage the consumption of carbohydrate-loaded fluids before surgery[4]. Carbohydrate loading practices vary among institutions, encompassing both simple carbohydrates (e.g. Gatorade) and complex carbohydrates (e.g., maltodextrin), with multiple commercial preparations available[5]. In the context of surgery, carbohydrate loading may reduce insulin resistance and improve recovery[6], although the exact mechanisms are unknown, and is confounded by numerous interventions (e.g. analgesia, early removal of catheters, thromboprophylaxis, etc[7].). Moreover, ERAS® pre-surgery drinks often contain high amounts of protein (up to 80 g/L). Interestingly, the benefit of high-protein diet before surgery has not been demonstrated. Inversely, high animal protein intake was positively associated with cardiovascular mortality[8].

Counterintuitively, decades of research have demonstrated that short-term preconditioning with dietary restriction (DR) from two days to one week increases resilience against stress, including IRI to the kidney, liver, brain, heart, and vein grafts[9–15]. DR can include either restriction of total energy intake (calorie restriction) or dilution of specific macronutrients in the diet, such as total protein, or specific amino acids like methionine, cysteine, or tryptophan[13,16,17]. While DR has been extensively used in preclinical models, achieving voluntary food restriction in humans remains challenging, even for a short period of time[18]. Thus, one of the goals in this research area is to identify simple interventions that are feasible in humans and can replicate the benefits of DR without the burden of restricting food intake. Consequently, many studies have focused on alternative nutritional strategies or on the molecular effectors that mediate the benefits of dietary restriction[19].

From a nutritional standpoint, many of the pleiotropic benefits of calorie restriction can be triggered by the restriction of dietary protein or single amino acids[20]. These benefits include resistance to hepatic IRI[14], improved recovery upon limb ischemia[13], increased energy expenditure[21], and improved glucose and lipid homeostasis, even during short-term interventions[22–24]. Protein restriction also extends lifespan, while high-fat and high-protein diets may accelerate aging and reduce lifespan[25]. The interaction between protein and carbohydrate intake is also important, as the isocaloric replacement of dietary protein with carbohydrates drives greater benefit than isocaloric replacement with fat[26,27]. Interestingly, the type of carbohydrate used to replace protein seems to have relatively little effect on the overall protein restriction response[28].

On a molecular level, protein and calorie restriction modulate pathways, including insulin/insulin-like growth factor-1 (IGF-1)[29,30], the redox-sensing transcription factor Nuclear Factor Erythroid 2-Related Factor (NRF2)[31] and hydrogen sulfide (H_2S) signaling[14]. One feature unique to protein restriction is increased fibroblast growth factor 21 (FGF21) production, which responds to starvation and protein/amino acid restriction, but not calorie restriction[32,33]. This is a conserved response, with circulating FGF21 increased 10-fold in rodents and approximately 2-fold in humans fed a low-protein (LP) diet for one month[32]. FGF21 signaling is associated with increased insulin sensitivity, improved glucose homeostasis, a reduction in circulating lipids, and an increased lifespan in mice[34–37]. FGF21 induces thermogenesis by increasing the expression of UCP1 in adipose tissue[38], which drives the browning of adipose tissue and increased energy expenditure[39]. During protein restriction, FGF21 signaling is essential to regulate adaptive homeostatic changes in metabolism and feeding behavior[40].

In this study, our aims were to 1) explore how *ad-libitum* access to carbohydrate-loading drinks impacts food intake, 2) evaluate the effects of one-week carbohydrate loading on IRI, a model of surgical stress, and 3) identify the mechanisms that underlie the benefits of carbohydrate loading. We hypothesized that mice with access to carbohydrate-loading drinks would consume less food, resulting in protein dilution. We further hypothesized that carbohydrate loading-induced short-term protein dilution would protect against surgical stress through conserved mechanisms of protein restriction.

We find that one week of carbohydrate loading reduces food intake and induces voluntary protein dilution in mice. Consistent with the benefits of low-protein diets[24,26], carbohydrate loading protects from surgical stress in two models: hepatic and renal IRI. In this context, FGF21 is required for protection from surgical stress, and FGF21 administration is sufficient to protect without any dietary intervention. Mechanistically, carbohydrate loading induction of FGF21 is associated with the induction of the canonical integrated stress response (ATF3/4, NF-kB), and oxidative metabolism (PPARγ).

## Results

### Short-term carbohydrate loading promotes resilience to surgical stress

Carbohydrate-loading drinks were modeled using a 50% (w/w) sucrose-water solution, hereafter referred to as "high carbohydrate - HC". The choice of sucrose as the source of carbohydrate was based on the following rationale: 1) Dietary protein restriction benefits have been achieved by substituting protein with sucrose[24,41]. 2) Metabolic outcomes of low (10%)-protein, high (70%)-carbohydrate diets are worse when carbohydrates comprise a mixture of monosaccharides fructose and glucose[28]. 3) Sucrose is commonly employed in ERAS protocols[5].

We began by examining the effects of HC prior to surgery by using a model of surgical stress that consisted of right nephrectomy and unilateral left kidney IRI (Supplementary Data Fig. 1a). Prior to IRI, 10-weeks-old C57BL/6 J male mice were given *ad libitum* access to water (Ctrl) or HC for one week. All mice were fed the same standardized *ad libitum* diet. All mice were given regular water postoperatively (Supplementary Data Fig. 1a). Mice on HC showed improved renal function after IRI, with reduced postoperative serum urea (Fig. 1a and Supplementary Data Fig. 1b, 935 ± 322 versus 1426 ± 309 AUC), day 3 serum creatinine levels (Fig. 1b, 0.4 ± 0.3 versus 1.11 ± 0.6 ng/mL) and improved glomerular filtration rate (GFR) as estimated by clearance of FITC-sinistrin (Supplementary Data Fig. 1c). HC also alleviated IRI as measured by histological assessment of tissue necrosis (Fig. 1c, 3.9 ± 0.94 versus 6.16 ± 1.8% necrotic area), lipid peroxidation (Supplementary Data Fig. 1d; 4HNE staining), and *Krt20* gene expression (Fig. 1d), a marker of proximal tubule damage[42].

Sex-specific differences have been observed in preclinical models of renal ischemic injury. In particular, females recover more readily from IRI than males[43]. Thus to substantiate our findings, we subjected C57BL/6 J female mice to the same HC preconditioning for one week, followed by renal IRI (Supplementary Data Fig. 1a). In accordance with what was observed in males, HC preconditioning reduced IRI in female mice (Fig. 1e and Supplementary Data Fig. 1e, serum urea 318 ± 47 versus 536 ± 79 AUC). Similarly, serum creatinine levels (Fig. 1f, 0.28 ± 0.05 versus 0.49 ± 0.08 ng/mL), tissue necrosis (Fig. 1g, 0.92 ± 0.30 versus 2.63 ± 0.70% necrotic area), and *Krt20* expression (Fig. 1h) were reduced in HC female mice compared to Ctrl.

Some benefits of DR appear lost when started late-in-life[44], but most surgical patients are older. Thus, we asked whether HC was similarly effective in aged male mice (22-months-old). As in young mice, HC preconditioning improved renal function upon IRI in old mice (Fig. 1i and Supplementary Data Fig. 1f, serum urea 593.3 ± 40.73 versus 733.8 ± 28.69 AUC), reduced renal necrosis (Figs. 1j, 4.6 ± 0.94 versus 7.69 ± 0.48%) and *Krt20* expression (Fig. 1k). Moreover, one

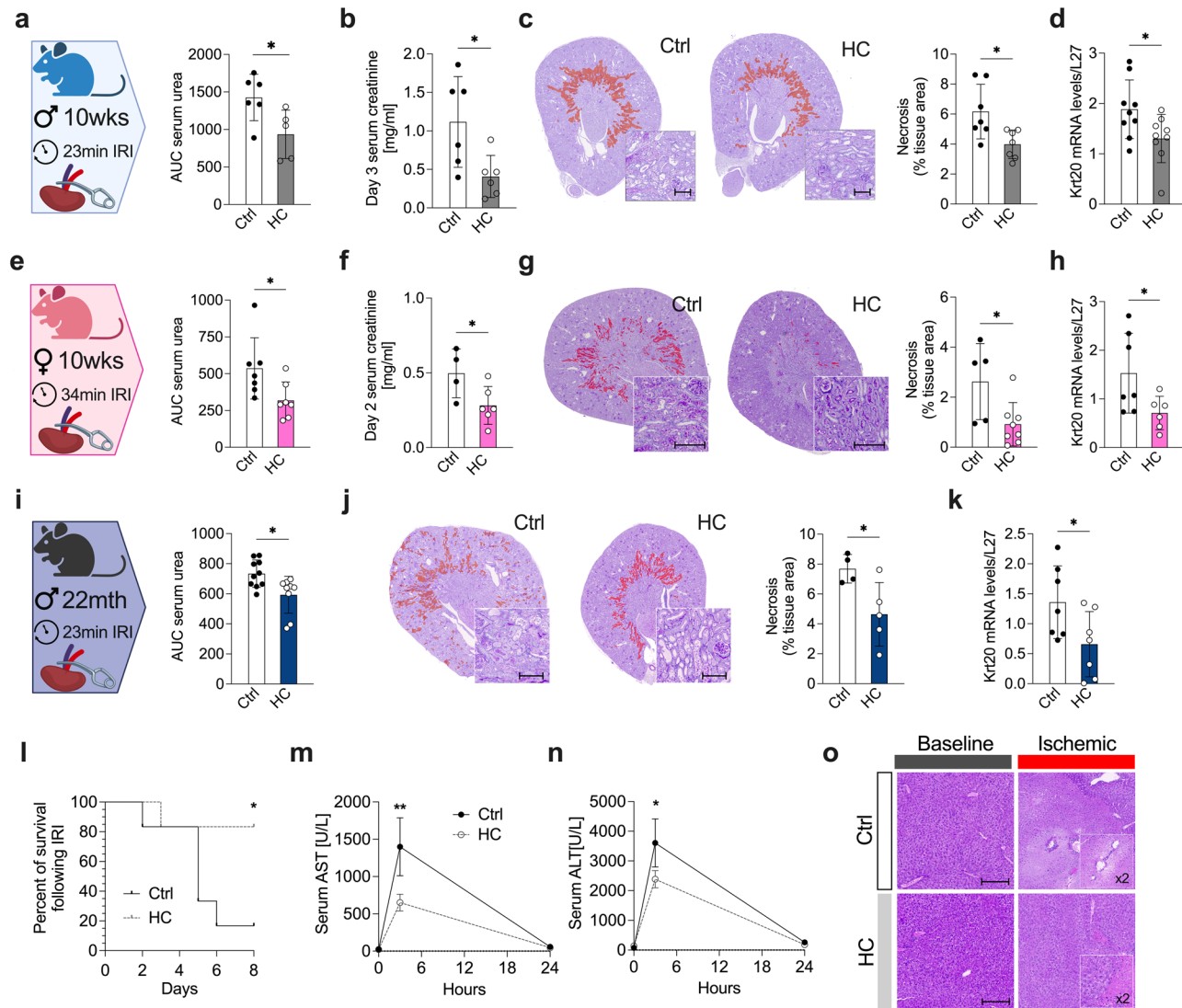

**Fig. 1 | Improved resistance to ischemia-reperfusion injury (IRI) following carbohydrate loading with 50% sucrose water. a** Experimental group (left) and serum urea levels (right; AUC) post-renal IRI after one week of preconditioning with Ctrl or HC. Created with BioRender.com. **b** Serum creatinine levels at day 3 post-renal IRI. **c**. Representative cross sections of PAS-stained kidneys with necrotic area digitally highlighted in red (left; x10 magnification; scale bar 100 μm) and necrotic tissue area quantification (right) at day 2 post-renal IRI. **d** Relative Krt20 mRNA levels in kidney at day 2 post-renal IRI. **e** Experimental group (left) and serum urea levels (AUC) post-renal IRI after one week of preconditioning with Ctrl or HC. Created with BioRender.com. **f** Serum creatinine levels at day 2 post-renal IRI. **g** Representative cross sections of PAS-stained kidneys with necrotic area digitally highlighted in red (left; x10 magnification; scale bar 100 μm) and quantification of necrotic tissue area (right) at day 2 post-renal IRI. **h** Relative Krt20 mRNA levels in kidney at day 2 post-renal IRI. **i** Experimental group (left) and serum urea levels (AUC) post-renal IRI after one week of preconditioning with Ctrl or HC. Created with BioRender.com. **j** Representative cross sections of PAS-stained kidneys with necrotic area digitally highlighted in red (left; x10 magnification; scale bar 100 μm) and quantification of necrotic tissue area (right) at day 2 post-renal IRI. **k** Relative Krt20 mRNA levels in kidney at day 2 post-renal IRI. **l** Kaplan-Meier survival curve

post-renal IRI after one week of preconditioning with Ctrl or HC. **m** Serum aspartate aminotransferase (AST) and (**n**) alanine aminotransferase (ALT) enzyme levels at the indicated time post-hepatic IRI. **o** Representative cross sections of H&E-stained livers (x20 magnification; scale bar 30 μm) at baseline or at day 1 pos-thepatic IRI. *$p$ values for **a-k** were calculated with unpaired two-tailed T-test, and for **l** with two-way repeated measures (RM) ANOVA with Geisser-Greenhouse correction and n, m with two-way ANOVA followed by a Sidak's post hoc analysis, *$p < 0.05$ **$p < 0.01$. $p$ values for **a** = 0.303, for **b** = 0.0234, for **c** = 0.0154, for **d** = 0.0345, for **e** = 0.0352, for **f** = 0.0467, for **g** = 0.0246, for **h** = 0.0445, for **i** = 0.0107, for **j** = , for **k** = 0.0427, for **l** = 0.0462, for **m** = 0.0346, for **n** = 0.0252. Images 1a, 1e and 1i were partly created with BioRender.com. ♂ indicates males, ♀ indicates females;. In **a-d** and **l-o**, experiments were carried out in 10-weeks old male mice; in e-h, in 10-week-old female mice; in **i-k**, in 22-month-old male mice. Sample sizes: (**a–c**), $n = 6$-7, $n = 7$ for Ctrl diet, and $n = 6$ for HC diet; (**d**) $n = 9$ in all experimental groups; (**e–h**), $n = 7$ in all experimental groups; (**i**), $n = 10$ in all experimental groups (2 experiments pooled, experiment 1: $n = 7$ in all experimental groups, experiment 2: $n = 3$ in all experimental groups; (**j-k**) $n = 7$ in all experimental groups; (l–o), $n = 6$ for all conditions. Data shown as mean ± SD. See also Supplementary Data Fig. 1.

week of pre-operative HC caused a significant survival advantage after severe (35 min) renal ischemia, with only 20% mortality at 8 days post-surgery compared to 80% mortality in the Ctrl group (Fig. 1l).

Finally, we tested whether HC could also protect from surgical stress by using a mouse model of hepatic IRI, as previously described[14]. Here, 10-weeks-old male mice on HC had decreased liver damage

relative to Ctrl at 3 hours post-surgery, as measured by serum aspartate aminotransferase (AST, Fig. 1m, 651 ± 275 versus 1400 ± 952 U/L) and serum alanine transaminase (ALT, Fig. 1n, 2384 ± 706 versus 3608 ± 1622 U/L). Histological analysis of hemorrhagic necrosis in livers collected 24 hours after reperfusion was consistent with liver damage markers and a protective effect of the HC (Fig. 1o). Taken

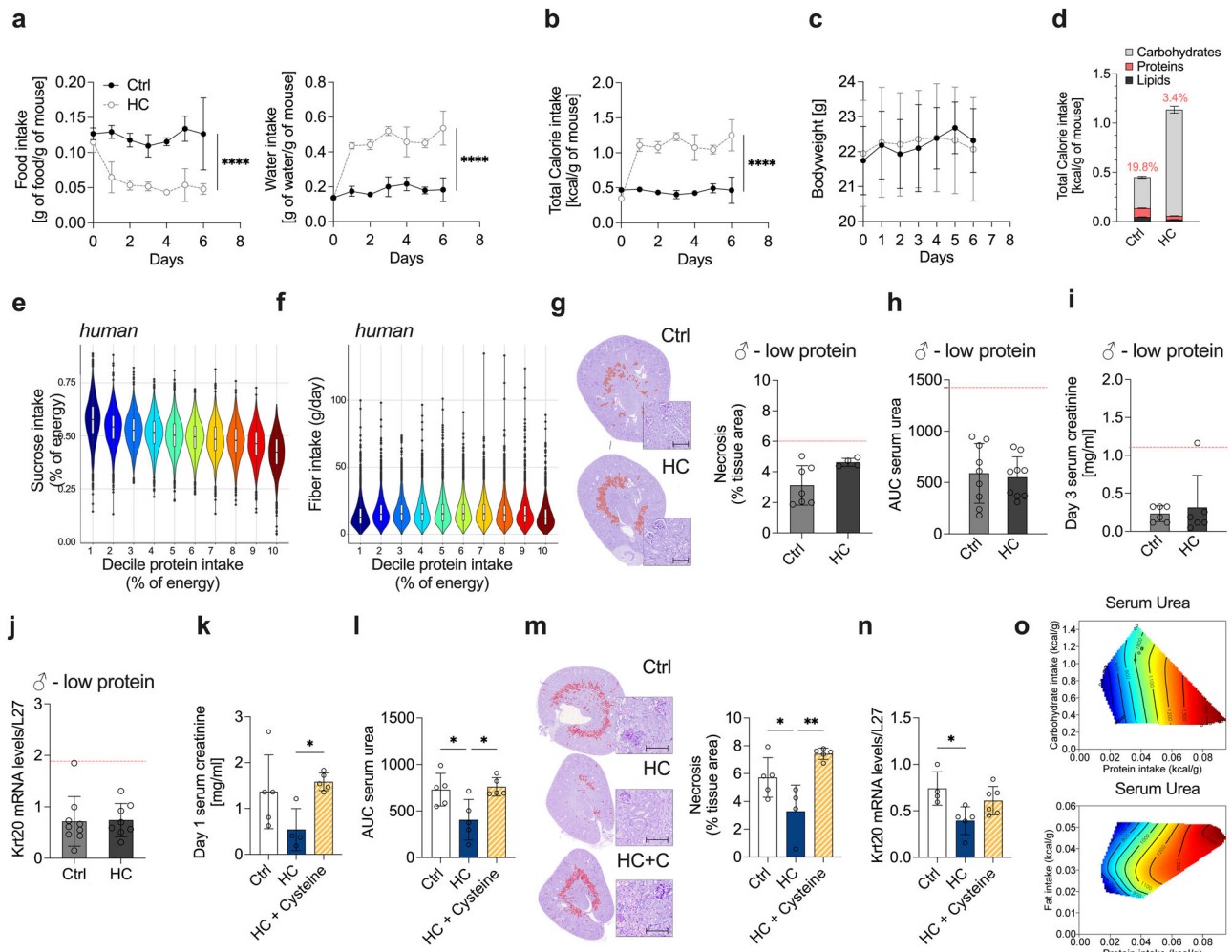

**Fig. 2 | Carbohydrate loading-induced protein dilution is required for protection against IRI. a** Food (left) and water (right) intake (normalized by body weight) of 10-weeks-old male mice given *ad libitum* access to water (Ctrl) or 50% sucrose-water (HC). **b** Total calorie intake normalized to body weight by day. **c** Body weight at the indicated time in mice given *ad libitum* access to Ctrl or HC. **d** Total calorie intake by macronutrient normalized to body weight by day. The % of protein intake is indicated on top of each bar. **e** Relationship between sucrose intake and protein intake, and (**f**) fiber intake and protein intake in humans from the US National Health and Nutrition Examination Survey (NHANES). Box plots represent median with 25th and 75th percentile. Whiskers plots represent hinge ± 1.5 IQR. **g** Representative cross sections of PAS-stained kidneys with necrotic area digitally highlighted in red (left; x10 magnification; scale bar 100 µm) and quantification of necrotic tissue area (right) at day 2 post-renal IRI after one week of preconditioning with low protein diet (LP) combined with Ctrl or HC. The red line indicates the level of the control group (Ctrl). **h** Serum urea levels (AUC) post-renal IRI after one week of preconditioning with the indicated diet. The red line indicates the level of the control group (Ctrl). **i** Serum creatinine levels at day 3 post-renal IRI. The red line indicates the level of the control group (Ctrl). **j** Relative Krt20 mRNA levels in kidneys at day 2 post-renal IRI after one week of preconditioning with the indicated diet. The red line indicates the level of the control group (Ctrl). **k** Serum creatinine levels at day 1 post-renal IRI after one week of preconditioning with Ctrl, HC or HC with cysteine oral gavage (HC + cysteine). **l**. Serum urea levels (AUC) post-renal IRI after one week of preconditioning with the indicated diet. **m**. Representative cross sections of PAS-stained kidneys with necrotic area digitally highlighted in red (left; x10 magnification; scale bar 100 µm) and quantification of necrotic tissue area (right) at day 2 postrenal IRI after one week of preconditioning with the indicated diet. **n** Relative Krt20 mRNA levels in kidney at day 2 post-renal IRI after one week of preconditioning with the indicated diet. **o** Relationship between serum urea at day 2 post-renal IRI and dietary protein intake or carbohydrate or fat intake. In all surfaces, red indicates the highest value while blue indicates the lowest value, with the colors standardized across each slice. *p values for **a–c** were calculated with two-way repeated measures (RM) ANOVA with Geisser-Greenhouse correction, **k–n** with unpaired two-tailed T-test and **o**. with a generalized additive model (GAM; **k** = 6), *$p < 0.05$ **$p < 0.01$ ****$p < 0.0001$. p values for a are respectively <0.0001 and <0.0001, for b < 0.0001, for **k** = 0.0411, for **l** = 0.0283 and 0.0171, for **m** = 0.0474 and 0.0019, for **n** = 0.0174. In **a–d** and **g–o**, experiments were carried out in 10-weeks old male mice. Sample sizes: (**a–d**), n = 8; (**e-f**), n = 11064 men, 12181 women (g-j), n = 8-9, n = 8 for low protein Ctrl, and n = 9 for HC; (**k–n**), n = 5 for all conditions; (**o**), n = 16 for all conditions. Data in all panels are shown as mean ± SD. See also Supplementary Data Figs. 2 and 3.

together, these data suggest that pre-operative HC confers hepatic and renal protection against surgical stress, independent of sex or age.

## One-week of carbohydrate loading-induced voluntary protein restriction required for surgical stress resistance

Having demonstrated that one week of carbohydrate loading increased surgical stress resistance, we then sought to identify the underlying nutritional mechanism. Given the links between dietary restriction and protection from surgical stress[14], we hypothesized that

carbohydrate loading would impact food intake. Indeed, HC induced a voluntary 2-fold reduction in solid food intake (Fig. 2a, 0.05 versus 0.12 g food/g body weight), with effects apparent within 24 hours. Strikingly, mice on HC doubled their water intake (Fig. 2a, 0.42 versus 0.18 g water/g body weight). Changes in the HC group drove a significant increase in total energy intake relative to body weight (Figs. 2b, 1.13 ± 0.09 versus 0.45 ± 0.03 kcal/g body weight). One week of HC did not affect bodyweight, lean or fat mass percentages compared to Ctrl (Fig. 2c and Supplementary Data Fig. 2a). Moreover, spontaneous

activity level was unchanged, as measured by average walk time over 72 hours (Supplementary Data Fig. 2b). When examining the contribution of each macronutrient to whole body energy intake, HC resulted in a significant increase in energy from carbohydrates (Figs. 2d, 1.08 ± 0.08 versus 0.31 ± 0.02 kcal/g body weight), and a decrease in energy from protein (Fig. 2d, 0.04 ± 0.005 versus 0.09 ± 0.007 kcal/g body weight) and fat (Fig. 2d, 0.02 ± 0.003 versus 0.05 ± 0.004 kcal/g body weight). In both sexes and aged male mice, HC induced a reduction in food intake, associated with a 2-fold increase in sucrose-water intake (Supplementary Data Figs. 2c and 2f). Similarly, body weight was unaffected (Supplementary Data Fig. 2d, g), and HC increased total energy intake relative to body weight (Supplementary Data Fig. 2e, f).

Given the robustness of these data, we asked whether similar effects could be observed in humans. Total carbohydrate and protein intake was analyzed from the National Health and Nutrition Examination Survey (NHANES) from 2005 to 2012 including two 24-h dietary recalls per person on a representative sample of the United States population every 2 years (four distinct cycles of data collection). Using this large dataset (n = 23,245), we indeed found an inverse association between sucrose and protein intake (Fig. 2e), but not between fiber and protein intake (Fig. 2f), consistent with our observations in rodents (Fig. 2a–d).

Given the voluntary protein restriction observed during carbohydrate loading, and because restricting dietary protein has pronounced effects on glucose tolerance[26], lipid metabolism[24] and resistance to IRI[12] in vivo, we hypothesized that HC might not have additional benefits in mice fed a low protein (LP) diet. To test this hypothesis, we fed an additional cohort of mice an LP diet (Supplementary Data Figs. 3a, 6.4 versus 19.8% of energy from protein in the standard diet), in the presence or absence of HC. Similar to mice on a standard diet, mice on an LP diet with access to HC decreased their food intake and increased their water and total energy intake (Supplementary Data Fig. 3B). Mice on an LP diet for one week lost weight (Supplementary Data Fig. 3c), but the LP diet did not affect total energy intake compared to a standard diet (Supplementary Data Fig. 3d). In line with our hypothesis that the benefits of carbohydrate loading are mediated by protein dilution, mice fed with an LP diet with access to HC had no additional protection from IRI, as assessed by tissue necrosis, serum urea, creatinine levels, and *Krt20* expression (Fig. 2g–j).

Next, we asked whether the reintroduction of protein would reverse the benefits of HC. To achieve this, we administered cysteine, a sulfur-containing amino acid previously shown to be relevant in the beneficial effects of DR[14], via oral gavage to mice treated with HC. We estimated the amount of cysteine to be 5.58 mg/day per mouse, which corresponded to the cysteine intake deficit in HC versus Ctrl mice. Interestingly, food, water, total energy intake, and body weight were unaffected by cysteine add-back (Supplementary Data Fig. 3e, g, HC + C). On the other hand, consistent with our hypothesis, the addition of cysteine to the diet (HC + Cysteine) abrogated HC benefits on surgical stress. This was evident by increased serum creatinine at day 1 and serum urea levels after IRI (Fig. 2k, l and Supplementary Data Fig. 3h) in the HC + Cysteine group. In addition, tissue necrosis and *Krt20* expression in the HC + Cysteine mice were comparable to those observed in the Ctrl (Fig. 2m, n). Finally, we utilized the geometric framework[26] to evaluate the effects of *ad libitum*-fed diets varying in macronutrients on kidney function (serum urea). Resistance to surgical stress was most robustly associated with protein intake (Fig. 2o). In contrast, fat and carbohydrate intake had negligible influence. Interestingly, while HC induced a further reduction in protein energy intake in the LP diet group (LP + HC 1.7%, LP + Ctrl 6.4%), no additional benefits on IRI protection were observed, possibly associated with a specific threshold, rather than a "linear effect" as previously observed[24,41]. Overall, we found that one week of HC increased total caloric intake while reducing protein intake, which is required for resistance to surgical stress.

## Carbohydrate loading induces FGF21 signaling

Nutrient sensing pathways proposed to link dietary restriction with stress resistance include $H_2S$ production downstream of the integrated stress response (Activating Transcription Factor 4; ATF4) and oxidative stress responses (NRF2). We previously demonstrated that $H_2S$ was necessary and sufficient for increased stress resistance under certain DR paradigms[13,14]. Surprisingly, HC in both standard and LP diets did not induce the $H_2S$-producing enzyme cystathionine-γ-lyase (CGL; *Cth* gene) in either the liver or kidney (Supplementary Data Fig. 4a). Similarly, $H_2S$ production capacity in the liver, kidney, and serum was unaffected by HC (Supplementary Data Fig. 4b, d). To experimentally validate a lack of involvement of $H_2S$ production in HC protection, mice were administered propargylglycine and aminooxyacetic acid (PAG + AOAA) to inhibit CGL-dependent $H_2S$ production (Supplementary Data Fig. 4e). We fed standard or LP diets to wild-type mice with access to Ctrl or HC drinks, and treated them with either vehicle or PAG + AOAA. Resistance to IRI, assessed by serum urea and creatinine levels, was significantly improved by HC, LP, and LP + HC in both vehicle- and PAG-treated mice (Supplementary Data Fig. 4f, g) compared to Ctrl mice, confirming that $H_2S$ production was not required for the protection induced by HC.

Fibroblast growth factor 21 (FGF21) is another key nutrient-sensing molecule[32]. In both rodents and humans, one month of protein restriction increases FGF21 and drives key physiologic adaptations, including thermogenesis and altered feeding behavior[32,37,45]. Here, one week of HC induced a 13-fold increase in serum FGF21 levels (Fig. 3a), which was already observed after 2 days on HC (Supplementary Data Fig. 5a). HC induced a similar increase in serum FGF21 levels in females (Supplementary Data Fig. 5b) and in old mice (Supplementary Data Fig. 5c). The LP diet increased serum FGF21 levels to a greater extent, but the combination of HC water and the LP diet did not result in an additional increase (Fig. 3a). FGF21 is a metabolic hormone predominantly produced in the liver[32]. Here Fgf21 mRNA expression was restricted to the liver, with low expression in the kidney, skeletal muscle, and adipose tissue (Supplementary Data Fig. 5d). Consistently, FGF21 induction was observed in the liver (Fig. 3b), but not in the kidney (Fig. 3c) of HC mice. In accordance with the activation of FGF21, HC increased whole-body metabolic rate, as shown by $VO_2$ consumption (Supplementary Data Fig. 5e) and $VCO_2$ production (Supplementary Data Fig. 5f), as well as energy expenditure (Supplementary Data Fig. 5g). Similarly, the effects of HC were driven by protein dilution, as no additive effect was observed in mice fed an LP diet (Supplementary Data Fig. 5e, g). Underlying these changes in metabolism, the HC diet-induced subcutaneous adipocyte browning with smaller multilocular adipocytes on H&E staining (Supplementary Data Fig. 5h) and a 2-fold increase of the thermogenesis-associated gene *Ucp1* in brown and white adipose tissue (Supplementary Data Fig. 5i, j). Serum FGF21 levels correlated negatively with dietary protein intake (Fig. 3d, r2 = 0.43, p = 0.0001) and with kidney function at day 2 post-renal IRI as measured by serum urea AUC (Fig. 3e, r2 = 0.30, p = 0.0167). Cysteine supplementation (HC + Cysteine) also reduced FGF21 (Fig. 3f), which is consistent with our hypothesis that protein dilution is required for the HC-mediated benefits.

To further explore the importance of HC-mediated induction of FGF21 signaling, we sought to compare the Fgf21 transcriptional signature to one week of HC. To do so, we examined the hepatic differentially expressed genes (DEGs) from a publicly available dataset of Fgf21-overexpressing mice (GSE39313)[35]. We identified 31 significant DEGs with an FDR p < 0.1 and log2FC > 1, overlapping with the HC transcriptional signature (Fig. 3g). Of these 31 DEG, the top 10 highest from the Fgf21-overexpression dataset were similarly upregulated in the livers of HC mice (Fig. 3h). Together, these data support the hypothesis that HC effects on stress resistance are mediated by FGF21 signaling induced downstream of dietary protein dilution.

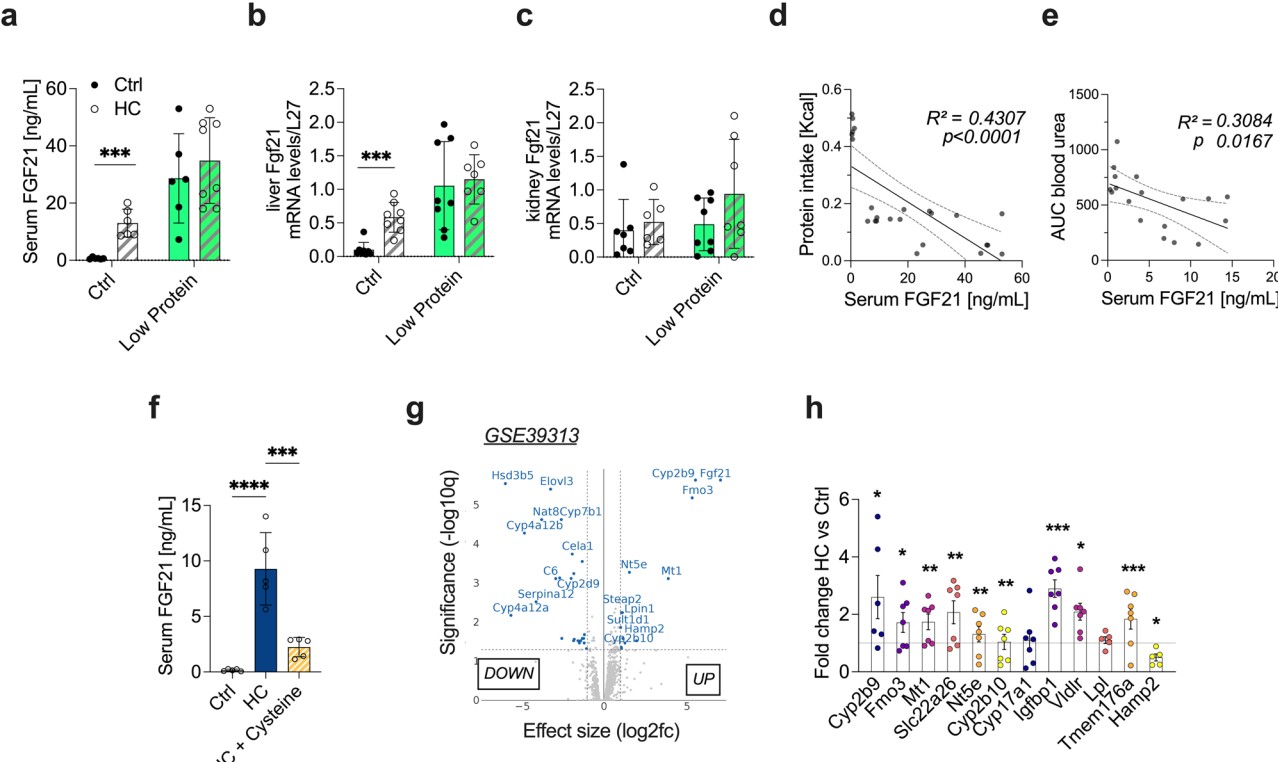

**Fig. 3 | Carbohydrate loading induces FGF21 expression. a** Serum FGF21 levels after one week of preconditioning with control or low protein diet (LP) with control (Ctrl) or 50% sucrose-water (HC). **b** Relative liver Fgf21 mRNA levels after one week of preconditioning with the indicated diet. **c** Relative kidney Fgf21 mRNA levels after one week of preconditioning with the indicated diet. **d** Linear regression showing the relationship between serum FGF21 levels versus protein intake and (e) serum urea (AUC). R² coefficient was calculated using Pearson's method. **f** Serum FGF21 levels after one week of preconditioning with Ctrl, HC or HC with cysteine oral gavage (HC + Cysteine). **g** Volcano plot of liver differential expressed genes (DEG) with FDR < 0.05 and log2FC > 2 from FGF21 overexpressing male mice from GSE39313. **h** Fold change of top DEGs from GSE39313 in liver after one week of preconditioning with HC compared to Ctrl. *p values of **a,b** were calculated with two-way ANOVA followed by a Tukey's post hoc analysis, **d,e** are linear regressions and r squared Pearson's coefficient, f with one-way ANOVA and h with unpaired two-tailed T-test with Bonferroni correction, *$p < 0.05$ **$p < 0.01$ ***$p < 0.001$ ****$p < 0.0001$. p-values for **a** = 0.0001, for **b** = 0.0001, for **d** < 0.0001, for **f** < 0.0001, for **h** = 0.0003. In all panels, experiments were carried out in 10-weeks old male mice. Sample sizes: (**a–c**), n = 8; (**d-e**), n = 32; (**f**), n = 5 for all conditions; (**h**), n = 8 for all conditions. Data in all panels are shown as mean ± SD. See also Supplementary Data Figs. 4 and 5.

## FGF21 is required for the protection from IRI

The necessity of FGF21 for HC-mediated benefits was tested with whole-body Fgf21 deletion (Fgf21$^{KO}$) mice (Fig. 4a). HC-mediated changes in body weight (Supplementary Data Fig. 6a), total calorie intake (Supplementary Data Fig. 6b, c), food and water intake (Fig. 4b) were similar in both Fgf21$^{WT}$ and Fgf21$^{KO}$ mice. However, while the absence of FGF21 had no effect on renal IRI response in Ctrl mice, Fgf21$^{KO}$ mice failed to gain protection upon HC, as demonstrated by serum urea (Fig. 4c and Supplementary Data Fig. 6d), creatinine levels (Fig. 4d), histological damage (Fig. 4e), tissue necrosis (Fig. 4f), and *Krt20* gene expression (Fig. 4g). Finally, we tested whether protection against general surgical stress conferred by HC required FGF21. Indeed, HC-mediated protection against hepatic IRI was also lost in Fgf21$^{KO}$ mice, as demonstrated by serum AST / ALT levels (Fig. 4h) and histological damage (Fig. 4i). In addition, *Hmgb1* mRNA expression, a biomarker of cellular stress responses and damage-associated molecular patterns[46], was decreased in Fgf21$^{WT}$ mice on HC, but not in Fgf21$^{KO}$ mice (Fig. 4j). In conclusion, our findings demonstrate that FGF21 is required for surgical stress resistance via mechanisms that may be independent of diet-induced metabolic adaptations.

## Exogenous FGF21 is sufficient to promote stress resilience

Long-term treatment with FGF21 agonists improves circulating lipid profiles in humans[47], but could negatively impact blood pressure with long-term use[48]. After showing that FGF21 was required for

protection against IRI, we tested whether short-term FGF21 treatment was sufficient to confer protection against surgical stress. Mouse FGF21 (mFGF21) at 1 mg/kg/day or sodium chloride vehicle (Veh) was administered continuously for one week via osmotic minipumps prior to renal IRI surgery (Fig. 5a). Serum FGF21 levels in mice treated with FGF21 were significantly higher than in mice given access to HC water (100 ng/mL versus 0.48 ng/mL, Fig. 5b). During the baseline period prior to surgery, and consistent with previous findings in Fgf21 transgenic mice[35], total food intake and water intake doubled in mice treated with FGF21 (Fig. 5c). On day 7 of treatment, FGF21-treated mice had a lower body weight compared to the control group (Supplementary Data Fig. 6e), which can be likely attributed to increased energy expenditure. Similar trends were observed for protein, carbohydrate, and lipid intake (Supplementary Data Fig. 6f, g). Immediately following renal IRI, and despite osmotic minipump removal, serum FGF21 remained high in the treated mice, but began decreasing by 48 hours post-surgery (Fig. 5b). Serum urea was significantly decreased in FGF21-treated mice compared to Veh (Fig. 5d, 530 ± 319 versus 813 ± 100, AUC). Serum creatinine at day 2 post-renal IRI was similarly reduced (Fig. 5e, 1.2 ± 1.3 mg/mL versus 3.5 ± 0.93 mg/mL). FGF21 preconditioning also decreased the expression of *Krt20* (Fig. 5f), decreased histological damage (Fig. 5g, 9.1 ± 3.4 versus 12.8 ± 2.9), and necrotic area (Fig. 5h). Together, these data are consistent with the hypothesis that FGF21 treatment for one week is sufficient to recapitulate HC-induced surgical stress protection.

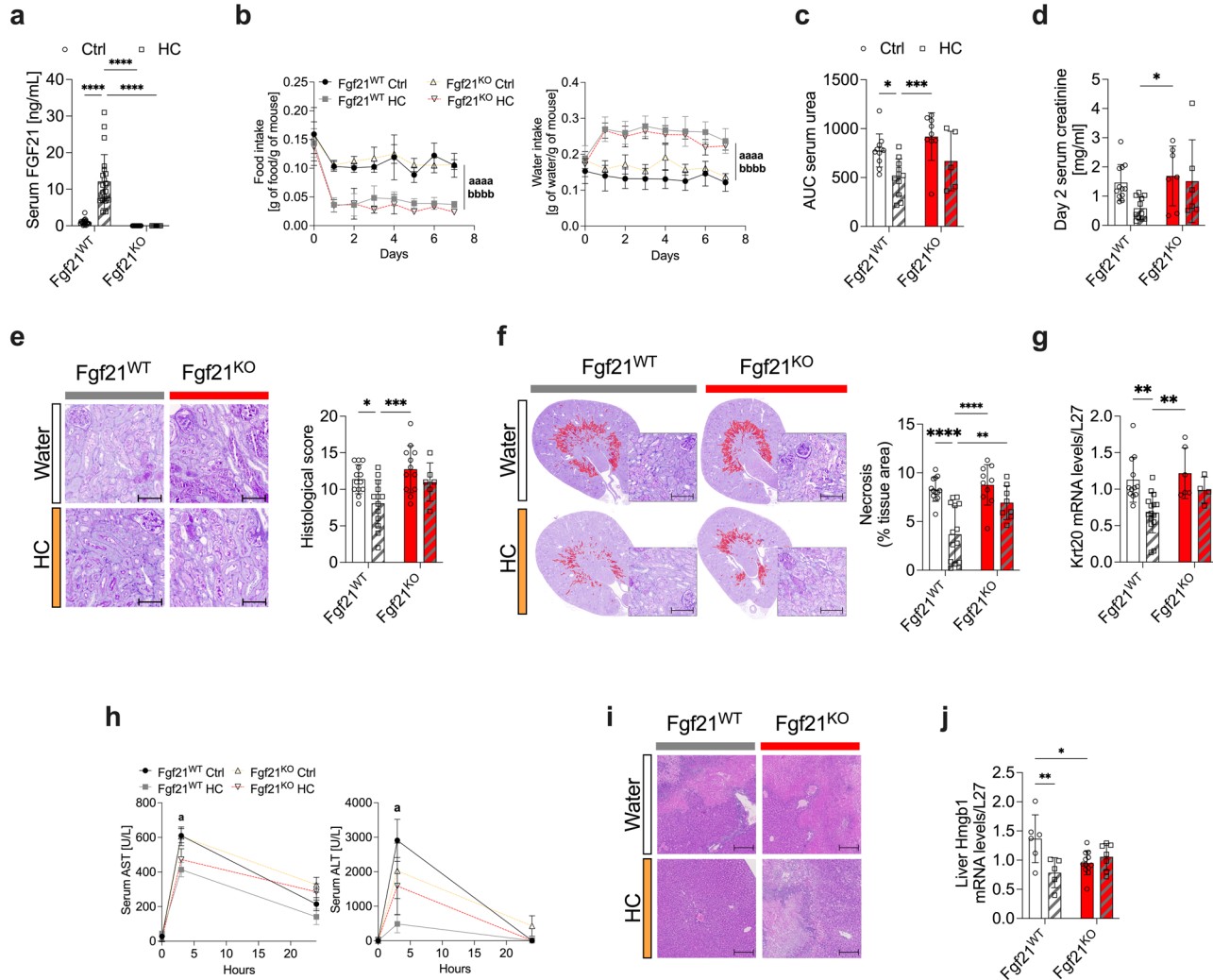

**Fig. 4 | FGF21 is required for the protective effects of carbohydrate loading.**
**a** Serum FGF21 levels after one week of preconditioning with standard (Ctrl) or 50% sucrose-water (HC). **b** Food (left) and water (right) intake (normalized by body weight) after one week of preconditioning with the indicated diet. **c** Serum urea (AUC) and (**d**) serum creatinine levels at day 2 post-renal IRI after one week of preconditioning with the indicated diet. **e** Representative cross sections of PAS-stained kidneys (left; x10 magnification; scale bar 100 μm) and histological score (right) at day 2 post-renal IRI after one week of preconditioning with the indicated diet. **f** Representative cross sections of PAS-stained kidneys with necrotic area digitally highlighted in red (left; x10 magnification; scale bar 100 μm) and quantification of necrotic tissue area (right) at day 2 post-renal IRI after one week of preconditioning with the indicated diet. **g** Relative Krt20 mRNA levels in the kidney at day 2 post-renal IRI after one week of preconditioning with the indicated diet. **h** Serum aspartate aminotransferase (AST) and alanine aminotransferase (ALT) enzyme levels at the indicated time post-hepatic IRI. **i** Representative cross sections of H&E-stained livers (x20 magnification; scale bar 30 μm) at day 1 post-hepatic IRI.

Ischemic areas appearing in light pink. **j** Relative Hmgb1 mRNA levels in the liver at day 2 post-renal IRI after one week of preconditioning with the indicated diet. *p values for a,c-g,j were calculated with two-way ANOVA followed by a Tukey's post hoc analysis, b with two-way RM ANOVA with Geisser-Greenhouse correction and h with two-way ANOVA followed by a Sidak's post hoc analysis, *$p < 0.05$ **$p < 0.01$ ***$p < 0.001$ ****$p < 0.0001$ a $p < 0.05$ aaaa $p < 0.0001$ Fgf21WT HC vs. Fgf21WT Ctrl, bbbb $p < 0.0001$ Fgf21KO HC vs. Fgf21WT Ctrl. p values for a < 0.0001, <0.0001 and <0.0001, for **b** < 0.0001 and <0.0001, for **c** = 0.0269 and 0.0008, for **d** = 0.0453, for e = 0.0236 and 0.001, for **f** < 0.0001, <0.0001 and = 0.0074, for **g** = 0.0002 and 0.0031, for **h** = 0.0328 and 0.0351, for **j** = 0.007 and 0.022. In **a-j**, experiments were carried out in Fgf21WT and Fgf21KO10-weeks old male mice. Sample sizes: (**a–g**), n = 10-15, n = 15 for Fgf21WT Ctrl, $n = 15$ for Fgf21WT HC, $n = 15$ Fgf21KO Ctrl, $n = 10$ for Fgf21KO HC; (**h–j**), $n = 6$-13, $n = 6$ Fgf21WT Ctrl and HC, $n = 10$ for Fgf21KO Ctrl, $n = 13$ for Fgf21KO HC. Data in all panels are shown as mean ± SD. See also Supplementary Data Fig. 6.

## Identification of a carbohydrate loading and FGF21-dependant signature

We finally sought to understand the changes in gene expression induced by FGF21 during HC, and how they underlie the protective effects on IRI. To do this, we identified gene expression modules associated with serum FGF21 expression in both kidney and liver transcriptomic datasets using weighted correlation network analysis (WGCNA). For the kidney, we performed bulk RNA-seq in mice preconditioned for one week with either Ctrl or HC on both standard and LP diets. For the liver, we used a public dataset (SRA accession PRJNA851959) of male mice receiving *ad libitum* access to diets

containing different protein levels (0%, 2%, 6%, 10%, 14%, and 18% protein intake) for 7 days[41]. To identify the genes associated with various kidney and liver phenotypes and traits, all expressed protein-coding genes were aggregated and analyzed using WGCNA, including 13106 genes for kidney and 11880 for liver. A total of 13 and 15 WGCNA modules were identified (Supplementary Data Fig. 7a, 8a), with each WGCNA module containing 24 to 4693 genes. The maroon, pale-turquoise, and darkseagreen modules in the kidney, and the darkred, salmon4, darkorange, indianred, and navajowhite modules in the liver, correlated ($p < 0.05$) with circulating FGF21 levels (Fig. 6a, b). Genes in these modules significantly associated with serum FGF21 levels

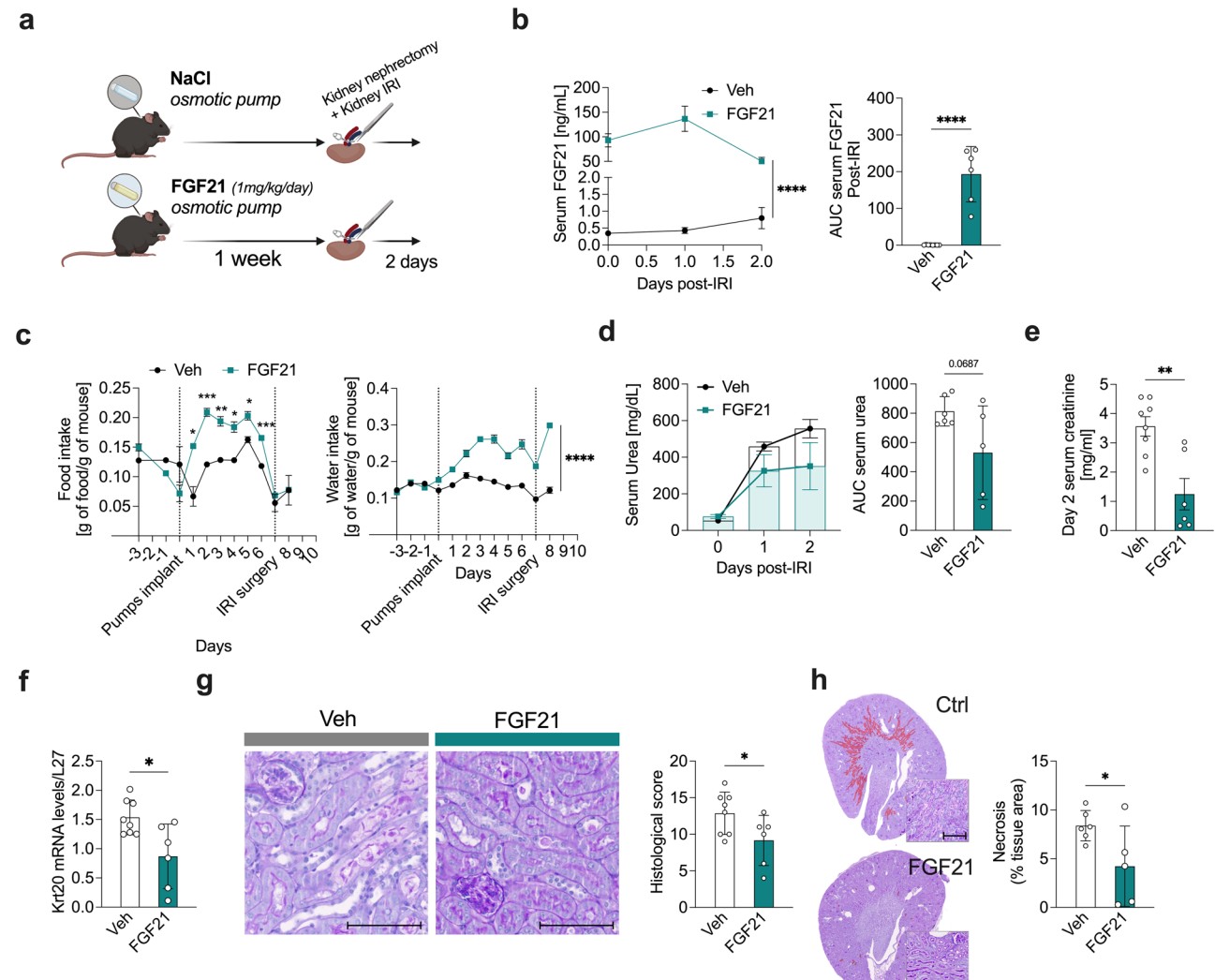

**Fig. 5 | FGF21 protects against kidney ischemia-reperfusion injury.**
**a** Experimental setup. Ad-libitum fed 10-weeks-old male mice were given free access to a standard diet (Ctrl) and implanted with osmotic pumps containing either NaCl (Veh) or human recombinant FGF21 (FGF21; 1 mg/kg/day) for 7 days prior to renal IRI surgery. Created with BioRender.com. **b** Serum FGF21 levels (left) and AUC (right) post-renal IRI after one week of preconditioning with the indicated treatment. **c** Food (left) and water (right) intake (normalized by body weight) after one week of preconditioning with the indicated treatment. **d** Serum urea levels (left) and AUC (right) at the indicated time post-renal IRI. **e** Serum creatinine levels at day 2 post-renal IRI after one week of preconditioning with the indicated treatment. **f** Relative Krt20 mRNA levels at day 2 post-renal IRI after one week of preconditioning with the indicated treatment. **g** Representative cross sections of PAS-stained kidneys (left; x10 magnification; scale bar 100 μm) at day 2 post-renal IRI and histological score (right) after one week of preconditioning with the indicated treatment. **h** Representative cross sections of PAS-stained kidneys with necrotic area digitally highlighted in red and quantification (left; x10 magnification; scale bar 100 μm) and necrotic tissue area (right) at day 2 post-renal IRI after one week of preconditioning with the indicated treatment. *p values for b,c were calculated with two-way RM ANOVA with Geisser-Greenhouse correction, c with two-way ANOVA followed by a Sidak's post hoc analysis, and b,d,e,f,g,h with unpaired two-tailed T-tests, *$p < 0.05$ **$p < 0.01$ ***$p < 0.001$ ****$p < 0.0001$. p values for **b** < 0.0001 and <0.0001, for **c** = 0.0120, 0.0002, 0.0028, 0.0121, 0.0129 and 0.0001, for **d** = 0.0667, for **e** = 0.0022, for **f** = 0.0129, for g = 0.0488, for **h** = 0.0471. Image1A was partially created with BioRender.com. In **b-h**, experiments were carried out in 10-week-old male mice. Sample sizes: (**a–h**), $n = 6$-8, $n = 8$ Ctrl and $n = 6$ FGF21 treated group. Data in all panels are shown as mean ± SD. See also Supplementary Data Fig. 6.

($R^2 > 0.5$, $p < 0.05$ in both liver and kidney) were involved in "ribonucleoprotein complex biogenesis", "ncRNA processing" and "regulation of apoptotic signaling pathway" (Fig. 6c). Genes inversely correlated with serum FGF21 levels were associated with metabolic processes, such as "regulation of cellular catabolic process" and "autophagy" (Fig. 6d and Supplementary Data Fig. 8b). Next, STRING (Search Tool for the Retrieval of Interacting Genes/Proteins) protein-protein interaction (PPI) network analysis was used to identify relevant transcription factors (TF) implicated in HC and FGF21-mediated benefits. E twenty-six (Ets), Nuclear factor kappa-light-chain-enhancer of activated B cells (Nf$_K$b), ATF4, Peroxisome proliferator-activated receptor gamma (PPARγ) and NRF2 (NES = 5.33, 10.26, 5.5, 9.4 and 6.78

respectively) were the most significant regulators associated with the binding motifs in the kidney module genes (Fig. 6e). Ets and Nf$_K$b have been implicated in the regulation of immune and cell survival processes, while PPARγ has been identified as a key regulator of cellular metabolism. Interestingly, in the liver, we identified ATF3/4, UQCRB, and Sox9 (NES = 13.49, 7.92 and 5.8 respectively) as the most significant regulators associated with binding motifs in genes positively correlated with serum FGF21 levels (Fig. 6f). ATF3 and ATF4 are involved in cellular stress responses, including amino acid deprivation and oxidative stress, while UQCRB is a subunit of complex III in the mitochondrial electron transport chain, suggesting the induction of tissue regeneration or repair-like responses. To validate our analysis,

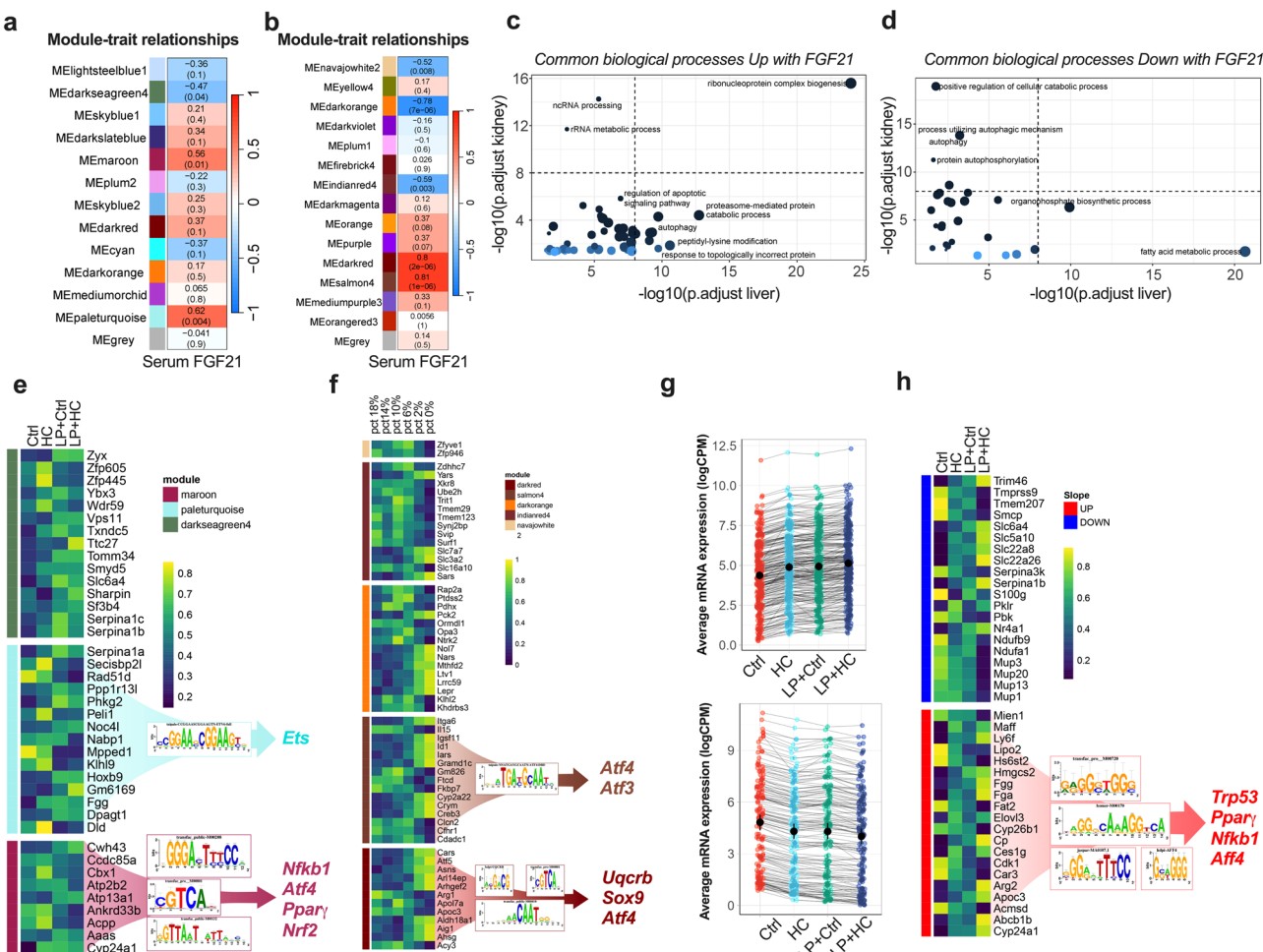

**Fig. 6 | Identification of FGF21-associated gene modules in kidney and liver induced by protein dilution. a** Module-Trait relationships with serum FGF21 levels identified through Weighted Gene Co-expression Network Analysis (WGCNA) from kidneys of male mice given access to *ad libitum* Ctrl, HC, LP and LPHC diet for one week. Top line corresponds to Pearson R² and bottom line to adjusted *p* value. Color scales represent positive correlation (red) and negative correlation (blue). **b** Module-Trait relationships with serum FGF21 levels identified through WGCNA from livers of mice after one week on diets with 18%, 14%, 10%, 6%, 2% and 0% protein content. Top line corresponds to Pearson R² and bottom line to adjusted *p* value. **c** Common biological processes (GO terms) enriched in kidney and liver modules with significant positive and (**d**) negative Module-Trait relationships with serum FGF21 identified through WGCNA. The dot size corresponds to the number of genes in the pathway. The color of the dots indicates the degree of significance of the pathway enrichment, with darker shades indicating higher significance. **e** Heatmap of percentile-transformed expression levels of the top 15 genes from kidney and liver (**f**) modules with significant Module-Trait relationships with serum

FGF21. Yellow indicating high expression and blue indicating low expression. The genes from each module with positive Module-Trait relationships with serum FGF21 were used in STRING and non-singleton were used for transcription factor analysis (TFA). The motifs picture shows the top NES motifs and their associated direct transcription factors. **g** Average mRNA expression levels of genes from kidney with slope > 10 and R² > 0.70 for each diet group (Ctrl, HC, LP, and LP+HC). Top panel shows genes with positive slopes, while bottom panel shows negative slopes, reflecting protein dilution intake. **h** Heatmap of percentile-transformed expression levels of the top 20 genes from kidney with slope > 10 and R squared > 0.70 for each diet group (Ctrl, HC, LP, and LP+HC). Yellow indicates high expression and blue indicates low expression. Upregulated genes were used in STRING and non-singleton were used for transcription factor analysis (TFA). The motifs picture shows the top NES motifs and their associated direct transcription factors. *p* values for a and b were calculated with Pearson correlation, and **c** and **d** with Benjamin-Hochberg (BH) test with FDR < 0.05. Sample sizes: (**g**), *n* = 3–6, *n* = 3 Ctrl, *n* = 3 HC, *n* = 6 LP, *n* = 5 LP + HC. See also Supplementary Data Fig. 7 and 8.

we employed an alternative method, identifying 263 positively (Fig. 6g top panel and Supplementary Data Fig. 8e) and 146 negatively (Fig. 6g bottom panel and Supplementary Data Fig. 8e) correlated genes with protein intake. In addition, network and transcription factor analysis identified PPARγ and Nf$_k$b, as well as Trp53 and ATF4 (Fig. 6h). Together, these results suggest that activation of stress responses and cell survival pathways is associated with FGF21 induction during HC.

## Discussion

In this study, we demonstrate that one week of carbohydrate loading induces protein dilution in mice, which confers protection from surgical stress in vivo via FGF21. We not only found that short-term carbohydrate loading is sufficient to induce protein dilution, but also

demonstrated that these benefits can be achieved under hypercaloric conditions. In fact, we are among the first to demonstrate a dietary intervention that protects from surgical stress without inducing body weight reduction. A model is presented in Supplementary Data Fig. 9.

Voluntary adherence to a dietary restriction is difficult for most people. One potential benefit of carbohydrate loading is that it does not involve forced food restriction but instead promotes an alternative source of energy and spontaneous macronutrient dilution. The response to carbohydrate was consistent across sex and in older mice, supporting the general validity of our findings. Interestingly, this differs from previous work in which low protein diet does not induce FGF21 in C57BL/6 J female mice[49]. Our carbohydrate loading protocol does induce greater protein restriction than diets used previously

(~ 3.5% vs 6% protein intake), so it is possible that females require greater protein restriction to induce FGF21. The observational human data from the USDA NHANES suggested an inverse association between sucrose and protein intake, indicating the potential for conserved appetitive mechanisms. However, the metabolic response and the extent to which carbohydrate loading may result in favorable outcomes in humans are still unknown. As with any epidemiologic study, the association of macronutrient intake can be confounded by any other variable (lifestyle, other dietary factors, comorbidities, medications, etc.). Although protein energy intake was treated as a single variable, it truly is a combination of amino acid ratios that cannot be fixed, even in mice which limit drawing conclusions about the effects of dietary protein in humans.

Here, carbohydrate loading consisted of an *ad libitum* solution of 50% sucrose in water for one week. This slightly differs from ERAS® protocols, classically recommending 100 g of carbohydrate the evening before surgery, and 50 g of carbohydrate until 2 hours before surgery[50]. Alternatively, healthy individuals have undergone 3-7 days of high carbohydrate loading diet (approximately 70% of calories from carbohydrate), at the expense of protein[51]. The latter is similar to most murine dietary protein restriction regimen, that replaced dietary protein with carbohydrates utilized complex carbohydrates derived from starch[15,41]. While we did not study the interactions between carbohydrate sources here, one of the few studies testing the effects of carbohydrate composition during protein dilution showed that a sucrose-rich diet improved metabolic parameters compared to a 50:50 mixture of glucose and fructose[28]. Of interest, carbohydrate loading resulted in an overconsumption of excess energy from sucrose and a voluntary reduction in protein intake. Rodents have a dominant protein appetite, which drives food intake to reach a protein target[52] via an FGF21-dependent mechanism[37,40]. Thus, protein-restricted diet typically results in the overconsumption of total energy. This response is called protein leverage[52]. A limited body of evidence also supports the protein leverage hypothesis in humans, at least down to 10% of protein energy[53]. While we do replicate the protein leverage hypothesis in mice fed low protein with normal water, we interestingly observe that mice given sucrose-water do not increase food intake on a low protein diet. This suggests the intriguing possibility that liquid carbohydrate intake may override protein leverage. Future studies may test the resulting hypothesis that FGF21 action on drinking behavior may override its action on feeding behavior, or that an FGF21-independent mechanism affects feeding behavior upon liquid carbohydrate intake.

Consistent with the latter hypothesis, carbohydrate loading-mediated changes in food and water intake were unaffected by the absence of FGF21. On the other hand, exogenous FGF21 administration, without any dietary intervention, was sufficient to increase both water and food intake. While protein restriction is required for the benefits of carbohydrate loading, this suggests that protein restriction might be dispensable for FGF21 benefits when given exogenously. These differences in the metabolic response to dietary protein restriction and protein dilution during carbohydrate loading will need to be further investigated. Additionally, liquid carbohydrate loading may be a previously unrecognized method to limit protein intake.

The response to dietary protein is also not linear. Previous work demonstrated that protein restriction has a threshold around 8-10% of dietary protein calories to produce the strongest improvements in body composition, insulin sensitivity[41] and induction of FGF21[24]. In our study, when comparing carbohydrate loading (3.4% dietary protein calories) versus a low protein diet (6.4% dietary protein calories), the protection from surgical stress was equivalent. This is consistent with the idea of a threshold and non-linear effects[41,54] A study comparing a range of sucrose concentrations during carbohydrate loading would be of interest to better investigate the potential for dose response to carbohydrate loading.

The duration of carbohydrate loading required for the onset of stress resistance benefits is unknown. Here, we show that only one week of carbohydrate loading is sufficient for the observed benefits in mice with maximal FGF21 induction already achieved after two days, indicating that this relatively short duration of carbohydrate loading might be sufficient to yield the observed benefits. Whether or not longer periods of carbohydrate loading will further increase protection remains to be experimentally determined.

Our previous results demonstrated that CGL is required for the benefits of DR, including protection from ischemia[13] and IRI[14]. Surprisingly, $H_2S$/CGL was not influenced by carbohydrate loading or protein restriction, nor required for surgical stress resistance. Protein restriction was associated with a decrease in CGL expression and cysteine level[41], while caloric and methionine restriction were previously demonstrated to increase both CGL and $H_2S$ production. CGL expression was also increased in response to a high-protein diet[55], but was not affected in long-lived mice overexpressing Fgf21[35,56]. Thus, our data demonstrate that, while CGL may be critical for the beneficial effects of calorie or methionine/cysteine restriction, it may be dispensable for the effects of carbohydrate loading/protein restriction in the liver and kidney.

Instead, FGF21 appears to be the molecular mediator of the benefits of carbohydrate loading. Additionally, we propose that HC-mediated protection from IRI functions as a para- and endocrine mechanism involving hepatic secretion of FGF21, consistent with a previous report[32]. This could serve as a basis for FGF21-based therapeutics during surgical stress or trauma, where there is currently no widely accepted risk mitigation strategy. FGF21 is a critical mediator of protein restriction[32]. In line with previous studies, we show that FGF21 signaling increases energy expenditure and thermogenesis[34,38,57]. During protein restriction, FGF21 signaling is essential to regulate adaptive, homeostatic changes in metabolism and feeding behavior[32,37]. Consistent with others[45], we found that carbohydrate loading induction of FGF21 caused increased adipose UCP-1 and increased energy expenditure. The importance of FGF21 signaling during carbohydrate loading also emphasizes that protein restriction is a key driver of the observed benefits. By which mechanism could endogenous FGF21 mediate the benefits of carbohydrate loading/protein restriction in mitigating IRI? While this remains unclear, our investigations into the regulatory networks and transcription factors involved in FGF21 benefits during carbohydrate loading suggest the importance of the stress response proteins ATF3/4 and the antioxidant regulator NRF2. Of note, in mice lacking ATF3, IRI was increased after orthotopic liver transplantation, which was associated with inhibition of HO-1 signaling and increased TLR4 as well as NFκB signaling[58]. ATF4 can induce hepatic FGF21 in both ATF4-overexpressing mice and a model of endoplasmic reticulum stress through eIF2alpha and the unfolded protein response[59]. However, the role of ATF4 in dietary restriction is complicated, as it is not required for FGF21 induction upon methionine restriction[60]. Similarly, NRF2 was not required for calorie restriction-dependent protection from hepatic IRI[14]. However, activation of NRF2 reduced tubular damage upon renal IRI and was associated with FGF21-mediated protection against diabetic nephropathy[61,62].

Current clinical recommendations for the clinical presurgical care are to 1) avoid fasting, 2) initiate nutritional support or supplementation without delay, and 3) reduce factors that exacerbate stress-related catabolism or impair gastrointestinal function[4,5]. Thus, our findings carry significant implications for the development of nutritional strategies during surgery. While our data suggest an inverse correlation between protein and sucrose intake in humans, the impact of short-term carbohydrate loading on calorie and protein intake in patients undergoing surgery remains to be established. Additionally, the impact of such interventions in humans with altered glucose metabolism (insulin resistance, metabolic syndrome, etc. will need to be

carefully evaluated prior to initiating a clinical trial. On the other hand, FGF21 could be an alternative strategy for those who are not able to tolerate carbohydrate loading[63]. Counterintuitively, FGF21 levels are increased in patients with obesity and chronic kidney disease (CKD)[64,65]. This may be explained by FGF21 resistance, which is also associated with worse metabolic profiles, higher inflammatory markers, more comorbidities, and higher mortality in CKD patients[66]. Thus, these patients may require either larger doses of exogenous FGF21 or an alternative intervention that promotes FGF21 sensitivity, such as carbohydrate loading/protein dilution, to achieve protection against IRI. Interestingly, in our study in FGF21-sensitive mice, short-term FGF21 delivery was sufficient to recapitulate the benefits of HC drinks without the need for prolonged administration, thus highlighting the translational potential.

In conclusion, in *ad libitum*-fed mice, carbohydrate loading promotes dietary protein restriction, the adaptive stress response, and resilience to surgical stress. ERAS® approaches currently promote short-term carbohydrate loading drinks to improve postoperative recovery without clear mechanistic explanations. Here, we identified FGF21 as a key molecular mediator of carbohydrate loading. These findings have broad implications for our basic understanding of the impact of carbohydrate and protein-carbohydrate interactions on metabolic health. Finally, our results provide a rationale for short-term carbohydrate loading or FGF21-based therapeutics during surgery, for which there is now no widely accepted risk mitigation strategy.

## Methods

### Mice
All experiments were performed with the approval of the cantonal Veterinary Office (Service de la Consommation et des Affaires Vétérinaires SCAV-EXPANIM, authorization number 3346, 3554b and 3768). All animal experimentation conformed to the *Guide for the Care and Use of Laboratory Animals*. All efforts were made to minimize animal suffering, including the use of anesthesia and analgesia during surgical procedures as well as humane endpoints to prevent undue distress. Animals were monitored daily. Mice with weigh loss > 10%, lethargy, muscle twitching, bleeding, infected or dehiscent wound were euthanized by exsanguination under general anesthesia in accordance with the *"Commission pour l'éthique dans les expérimentations animals"*. 10 to 12-weeks-old male and female, and 22-month-old male C57BL/6 J mice (Janvier Labs, France) were used for all experiments. All mice were kept on *ad-libitum* (AL) access to food and tap water, and kept under standard housing conditions, with 12 hr light/12 hr dark cycles, 30-70% humidity and a temperature of 20-23 °C unless specified otherwise. Fgf21 knockout (Fgf21^KO) was generated by crossing Fgf21^loxP mice (B6.129S6(SJL)-Fgf21^tm1.2Djm/J, Jackson Labs #022361) with loxP sites flanking exons 1-3 of the Fgf21 gene with CMV-cre expressing mice (B6.C-Tg(CMV-cre)1Cgn/J, Jackson Labs #006054). The resulting offspring had a deletion in exons 1-3 of Fgf21 in all tissues. The line was subsequently maintained by breeding animals heterozygous for the Fgf21 deletion allele. Mouse ear biopsies were taken and digested in DirectPCR lysis reagent with proteinase K. Wild type (WT), heterozygous and knockout (KO) mice were identified by PCR using the forward primer 5'1- ACC CCC TGA GCA TGG TAG A-3' to detect the WT allele, and forward primer 5'- CAG ACC AAG GAG CAC AGA CC-3' to detect the KO allele, and the common reverse primer 5'- GCA GAG GCA AGT GAT TTT GA-3', using GoTaqR G2 Green Master Mix (M7822, Promega).

### Experimental diets
All experimental diets were based on diet 2125 from Granovit AG (Kaiseraugst, Switzerland), with 19.8% of calories from protein (hydrolyzed casein and individual crystalline amino acids), 10.4% from fat and 69.9% from carbohydrate. The low protein (LP) diet was custom prepared by Granovit AG based on diet 2125 with 6.4% protein/ 10.4%

fat/ 83.2% of carbohydrate (Granovit AG) and was provided *ad-libitum*. The nutritional value of the control diet was 3.6 Kcal/gr, and 3.35 Kcal/gr for the LP diet. The high sucrose (HC) drink was prepared by adding 50% sucrose (w/w) in the drinking water, giving 2 Kcal of carbohydrate/g of HC drink. HC drink was changed every two days. Food pellets and bottles were weighed daily in every cage of mice. The delta weight of food and drink was calculated, divided by the number of mice, and normalized by the weight of each individual mouse in the cage. Total calorie intake was measured by calculating the sum of each macronutrient and their calories per gram (carbohydrates, proteins 4 kcal/gram, and fat 9 kcal/gram).

### Cysteine addback
C57BL/6 J 10 to 12 weeks-old male mice were gavaged with soft plastic oral 20 G gavage needles (Instech Laboratories). The cysteine solution concentration was calculated from the difference of daily food intake, in weight of food, between mice given *ad libitum* access to water (Ctrl) and HC. According to the diet composition, cysteine constitutes 0.30% of the food weight (diet 2125 from Granovit AG), resulting in a daily intake of cysteine of 0.00558 g/day. A stock solution containing a concentration of 27.9 mg/mL of L-Cysteine in drinkable water (#168149, Sigma) was prepared and stored at 4 °C. The prepared solution was administered via oral gavage (10uL/g of body weight) once daily (5.58 mg/day of L-Cysteine). Daily oral gavage of an equivalent volume of drinkable water (10uL/g body weight) was given as vehicle control.

### Nephrectomy and renal ischemia-reperfusion injury
Mice were anesthetized with 3% isoflurane in 2 L O$_2$ and kept at 37 °C with an electrical heating pad. Following a 2-cm abdominal incision, the vascular pedicles of the right and left kidneys were identified under a microscope. First, the right renal artery, vein, and ureter were ligated and cauterized. Immediately after, the right kidney was dissected and flash-frozen in liquid nitrogen. Second, the left pedicle was clamped for either 23 minutes for young male and old male mice or 34 min for young female mice with S&T vascular micro-clamps (FST 18055-03, Fine Science Tools). Of note, the survival experiment was performed as a bilateral clamping of both pedicles for 35 minutes. A darkening of the kidney was observed to ensure that the pedicle had been successfully clamped.

### Hepatic ischemia-reperfusion injury
Liver IRI was performed as described previously[14]. Briefly, mice were anesthetized with 3% isoflurane in 2 L O$_2$ and kept at 37 °C with an electrical heating pad. Following a 2-cm abdominal incision, the vascular pedicles of the median lobe and left lateral lobe were identified under a microscope. An atraumatic S&T vascular micro-clamp (FST 18055-03, Fine Science Tools) was placed across the portal vein, hepatic artery, and bile duct just above the branching to the right lateral lobe for 35 minutes. Blanching of the lobes was observed to ensure that the artery had been successfully clamped. After 35 minutes of ischemia, the clamp was removed.

### Postoperative management
The abdominal incision was sutured with 6-0 Prolene, and surgical staples were used to close the skin (427631, Aichele Medico AG). All mice received subcutaneous buprenorphine (0.05 mg/kg Temgesic, Reckitt Benckiser AG, Switzerland) 15 minutes before surgery and −6-8 hours following surgery. During the night, mice received paracetamol (2 mg/ml Dafalgan, UPSA) and buprenorphine (0.009 mg/ml Temgesic, Reckitt Benckiser AG, Switzerland) in the drinking water for 48 hours postoperatively. During the day, mice received subcutaneous buprenorphine injection in the morning and evening for 48 hours postoperatively. Blood samples from the tail vein were taken preoperatively, and at 3 h and 24 h postoperatively using a capillary tube.

Serum was separated by centrifugation (2000g for 20 minutes at 4 °C) and flash-frozen in liquid nitrogen before being stored at −80 °C. On the second- or third-day following surgery for renal IRI and 24 h following surgery for hepatic IRI, mice were euthanized under anesthesia via cervical dislocation, followed by exsanguination, and perfused with PBS. Of note, the surviving mice involved in the survival experiment were euthanized after 8 days. The remaining kidney was collected and cut in half transversally. One half was flash-frozen in liquid nitrogen, and the other was fixed in 10% neutral buffered formalin and paraffin-embedded for histology.

### FGF21 treatment

Mouse recombinant FGF21 (Cat# 450-56, Peprotech) was dissolved and diluted in sterile distilled water to a final dosage of 1 mg/kg/day. The filled 1007D Alzet osmotic minipump was presoaked for 24 hours in NaCl at 37 °C in a dry incubator. Mice were anesthetized with 3% isoflurane in 2 L O2 and kept at 37 °C with an electrical heating pad. A 1-cm incision was made in the skin of the upper back/neck to implant the sterile, preloaded minipump. 5-0 Prolene surgical thread was used to stitch the wound. Mice received paracetamol (2 mg/ml Dafalgan, UPSA) in the drinking water for 48 hours postoperatively.

### In-vivo inhibition of endogenous H₂S production

Inhibition of endogenous $H_2S$ production was achieved by injecting the mice with 100 µL of a stock solution of 1 mg/mL propargylglycine (PAG) and 1 mg/mL amino-oxyacetic acid (AOAA) in physiologic saline, or saline vehicle, as previously described[14]. Mice were injected intraperitoneally (i.p.) once a day for three days prior to the kidney IRI surgery.

### Metabolic cages

Throughout the calorimetry studies, a standard 12-hour light/dark cycle was maintained. Prior to data collection, all animals were weighed and acclimated to either normal chow or a low protein diet (LP) for the first three days, with or without a 50% sucrose drink. After 24 hours of acclimatization, mice were placed in metabolic cages, and measurements began for two consecutive days. At the conclusion of the measurements, the mice were weighed again. Energy expenditure was determined using a computer-controlled indirect calorimetry system (PromethionH, Sable Systems, Las Vegas, NV) as published[67]. Animals had unlimited access to food and water throughout the study. XYZ beam arrays (BXYZ-R, Sable Systems, Las Vegas, NV) were used to record ambulatory activity and position, and respiratory gases were measured using an integrated fuel cell oxygen analyzer, a spectrophotometric $CO_2$ analyzer, and a capacitive water vapor partial pressure analyzer (GA3, Sable Systems, Las Vegas, NV). Oxygen consumption and $CO_2$ production were monitored for 1-minute at 5-minute intervals. The respiratory quotient (RQ) was determined by dividing $CO_2$ production by $O_2$ consumption. The Weir equation was used to calculate energy expenditure: Kcal/hr = $60*(0.003941*VO_2 + 0.001106*VCO_2)$[68]. MetaScreen v. 2.5 was used to coordinate data acquisition and instrument control, and raw data was processed using ExpeData v. 1.8.5 (Sable Systems, Las Vegas, NV) via an analysis macro that detailed all aspects of data transformation (available on request from the corresponding author).

### Glomerular filtration rate measurement

FITC-sinistrin clearance was utilized to determine the glomerular filtration rate. Mice were anesthetized with isoflurane and subsequently had their right flanks shaved. After 2 minutes of basal recording, a mini-camera was connected to the flank, and a 0.35 g/kg solution of FITC-sinistin (Fresenius-Kebi) was injected intravenously into the tail vein. The mice were anesthetized again with isoflurane, and the camera was removed after 90 minutes of recording. MP&D lab software (Mannheim Pharma & Diagnostics) was used to analyze the FITC-sinistrin

data, and GFR was estimated as published[69]. The GFR was then normalized to control and represented in mL/min/kg of body weight.

### Body composition (EchoMRI)

Body composition (EchoMRI-100H, Echo Medical System, Houston, TX) was obtained on awake mice contained in a thin-walled plastic cylinder with a cylindrical plastic insert to restrict their movement. Mice were exposed to a low-intensity electromagnetic field for a brief period, and their fat and lean mass were determined.

### Histological analysis

3-micron sections from paraffin-embedded half kidneys were stained with PAS (Periodic Acid Schiff) and paraffin-embedded livers with H&E. The kidneys were scored histologically using a modified Goujon scoring method[70–72]. This score was created to limit the observer's subjectivity and to evaluate the entire section containing heterogeneous damage. The kidneys were imaged at a magnification of 20x using a Zeiss Axioscan Z.1 slide scanner (Carl Zeiss). The entire scanned section was analyzed using Zen Blue 3.4 software (Carl Zeiss) and scored on five parameters: 1) glomeruli integrity, 2) tubule dilatation, 3) brush border integrity, 4) debris in the tubules, and 5) medulla integrity in the cortico-medullar area. Within each category, each item was graded on a scale of 0 to 3, with 0 representing "no damage" and 3 representing "extremely damaged." Briefly, to assess glomerulus integrity, more than ten glomeruli were randomly selected from the section and assigned a score of 0 to 3 on a scale of 0 to 3. The same procedure was followed in the remaining categories. After that, the score for each category was converted to a percentage of damage. A final score between 0 and 5 was assigned based on this percentage of damage: 0 represents a percentage of damage between 0% and 15%; 1 represents a percentage of damage between 15% and 30%; 2 represents a percentage of damage between 30% and 45 percent; 3 represents a percentage of damage between 45 percent and 60%; 4 represents a percentage of damage greater than 60%; and 5 represents a percentage of damage greater than 75%. On a scale of 0 to 25, the final score was the sum of the scores for each category.

### Necrosis quantification

PAS-stained Entire scanned sections of kidneys were exported (50%) using Zen Blue 3.4 software (Carl Zeiss). Necrotic areas were identified using the following criteria: tubules with large debris, large dilation, and tubular cell loss; tubules with cast formation; and tubule loss. The pelvis region has been excluded from quantification.

The necrotic area of digitally highlighted images was measured by calculating the highlighted area fraction using ImageJ (v1.54r) and defined as the necrotic area divided by the total kidney area.

### Immunohistochemistry staining

4HNE (MAB3249, R&D Systems) immunohistochemistry was performed on paraffin sections[73]. After rehydration and antigen retrieval (TRIS-EDTA buffer, pH 9, 1 minute in an electric pressure cooker autocuiser Instant Pot duo 60 under high pressure), immunostaining was performed using the EnVision®+ Dual Link System-HRP (DAB +) according to the manufacturer's instructions. Slides were further counterstained with hematoxylin. The positive immunostaining area was quantified using the Fiji (ImageJ 1.54r) software and normalized to the total area of the tissue by two independent observers blinded to the conditions.

### Reverse transcription-quantitative polymerase chain reaction (RT-qPCR)

Mouse RNA was isolated using the TriPureTM method (Roche, Switzerland) from 30-50 mg of kidney, liver, and muscle powder, followed by complementary DNA (cDNA) synthesis using the Prime Script RT reagent kit (Takara). cDNA samples were loaded into a 384-well plate

format (Applied Biosystems, ThermoFischer Scientific AG, Switzerland) using SYBR Green reagent–based PCR chemistry (10-l reaction volume containing specific forward and reverse primers). The quantitative PCR program was run on a ViiA 7 Real-Time PCR System, according to the manufacturer's recommendations (Applied Biosystems, ThermoFischer Scientific AG, Switzerland). RPL27 was chosen as the housekeeping gene[74]. Ct values for candidate and housekeeping genes were determined, and standard curves for each gene were calculated using serial dilutions. The relative standard curve method was used to determine the relative level of expression of genes. For gene encoding, the primers listed in supplementary Table S1 were used, and analysis was performed using the QuantStudioTM 1.3 software (ThermoFischer Scientific AG, Switzerland).

## Blood analysis

Serum was isolated from blood taken from the tail vein pre-operatively and on days 1, 2, and 3 postoperatively by centrifugation (2000g for 20 minutes at 4 °C). Triglyceride concentration was determined using a Sigma kit (Cat# TR0100). Urea was measured as published[12] and using the Jung colorimetric method using a reactive solution containing 100 mg/l o-Phthalaldehyde (32800, Serva), 300 mg/l N-(1-naphthyl) ethylenediamine dihydrochloride (222488, Sigma), 2.5 mol/l sulfuric acid, 2.5 g/l boric acid, and 0,003 % Brij 35. After 30 minutes in the dark and at room temperature, the response was measured at 505 nm with a Synergy Mx micro-plate reader (BioTek Instruments (Switzerland) GmbH). The serum creatinine level was determined using the mouse Creatinine Assay Kit (80350, Crystal Chem INC). Mouse FGF-21 (MF2100, R&D Systems) levels were determined using commercial ELISA kits using the manufacturer's recommended protocol. The concentrations of AST (TR70121) and ALT (TR71121) were determined according to the manufacturer's instructions (ThermoScientific, Middletown, VA).

## Lead acetate assay

$H_2S$ production was quantified in the kidney and liver, as previously published[13,14]. Briefly, tissue powder was homogenized in passive lysis buffer (PLB E1941, Promega) and the protein concentration was determined using the ThermoFisher PierceTM BCA Protein Assay Kit (Cat# 23227). In a 96-well plate covered with Whatmann paper soaked in 20 mM lead acetate, samples were diluted (80 μg of protein in 80 L of PBS) and combined with 20 L of reaction mix containing PBS, 1 mM Pyridoxal 5'-phosphate (PLP) (82870, Sigma), and 10 mM Cysteine (c7352, Sigma). The plate was then incubated in a dry incubator at 37 °C for 2–4 hours until lead sulfide darkening of the paper occurred.

## Response surfaces analysis

All data were analyzed in R v.4.2.2. Data involving response surfaces were analyzed using Generalized Additive Model (GAM) and previously described[26]. Briefly, GAMs with thin-plate splines were used to model the responses to the macronutrient composition of the diet. GAMs were fitted using the mgcv tool of the R programming language (v1.9.0). The effects of macronutrients were divided into main effects and interactions. The response included post-renal IRI serum urea levels.

## NHANES analysis

Data from the US Department of Agriculture National Health and Nutrition Examination Survey were analyzed as previously described[75]. Briefly, NHANES XPT data files were downloaded from the CDC NHANES website for surveys carried out from 2005-2012. The data were imported into R using the sasxport.get function from the Hmisc package. Individuals who were not pregnant, over the age of 18 years, and had two complete 24-hour dietary recalls were included. As no gender-differences were observed, genders were pooled for the final analysis presented in the manuscript. For nutrient analyses, values from each individual's two 24-hour recalls were averaged.

## RNA-sequencing processing and computation

Data preprocessing, statistical computation, and visualization were performed using the Omics Playground version v2.8.10[76]. Data preprocessing included filtering genes based on variance, expression across the samples, and missing values. Only protein-coding genes on non-sexual chromosomes were included in the analysis. Batch effects were identified by an F-test for the first three principal components. Batch correction was performed for explicit batch variables or unwanted covariates. Parameters with a correlation r > 0.3 with any of the variables of interest (i.e., the model parameters) were omitted from the regression. Correction was performed by regressing the covariates using the 'removeBatchEffect' function in the limma R/Bioconductor package. For gene-level testing and identification of differentially expressed genes (DEG), statistical significance was assessed using two independent statistical methods: voom and limma-no-trend. Only genes that were significant using both methods were included. Gene expression was normalized using logCPM normalization in the edgeR R/Bioconductor package. *For slope analysis*, normalized genes were correlated with the mean protein intake of each group (Ctrl, HC in both regular and low protein diets) to identify protein dilution-driven genes. 409 genes with a correlation r2 > 0.6 and a slope > 9 were selected. Bonferroni corrections were applied to multiple Pearson correlations.

## WGCNA analysis

Raw counts were normalized by performing a variance-stabilizing transformation using the DESeq2 package in R (v1.40.2,). Genes with counts <15 in more than 75% of samples were filtered out. The variance-stabilizing transformed gene expressions were subjected to WGCNA based on the WGCNA package in R (v1.72-1)[77]. WGCNA parameters were set as: no missing data expression; soft threshold = 7 (estimate value); adjacency = "signed hybrid"; TOMType = " signed"; merge cut height = 0.4. Function annotations of the genes were obtained using the org.Mm.eg.db package in R (v3.15). The correlation of eigengenes with external traits (FGF21 levels) was performed using 'bicor' biweight midcorrelation to obtain the most significant and robust associations. Intramodular analysis identified genes with high gene-module membership and gene-trait significance. The ClusterProfiler package in R (v4.8.2) and STRING v11.5 (Search Tool for the Retrieval of Interacting Genes) was used for functional enrichment. Protein-Protein Interactions network was performed with the module/cluster analysis using the Markov Cluster Algorithm (MCL) in Cytoscape[78].16,377 detected genes were used as the enrichment background. Terms with a False Discovery Rate (FDR) < 0.05 were included.

## Transcription factor analysis

Transcription factor binding motifs, which are enriched in the genomic regions of a query gene set, were determined using the iRegulon plugin (version 1.3) in Cytoscape. The cis-regulatory control elements of genes from each module associated with serum FGF21 were used in the iRegulon analysis. We set: identity between orthologous genes ≥ 0.01, FDR on motif similarity ≤ 0.001, and TF motifs with a normalized enrichment score (NES) > 3.5, ranking option for Motif collection :10 K (9713 PWMs), and a putative regulatory region of 20 kb centered around TSS (7 species). For gene expression analysis the logCPM normalized genes with non-missing values and a gene-module significance > 0.6 or < −0.6 with serum FGF21 levels, as well as the top positive and negative slopes from the gene expression-protein intake association, were selected. Gene expression of selected genes underwent a z-transformation, and the top genes from each module were

used for the heatmaps using pheatmap in R (v1.0.12). RNA sequencing publicly available data of liver of the Fgf21 transgenic mice (GSE39313)[35] and protein restriction diet (SRA accession PRJNA851959)[41] were analyzed following the same pipeline.

**Quantification, statistical analysis and artwork**
All experiments adhered to the ARRIVE guidelines and followed strict randomization. All experiments and data analysis were conducted in a blind manner using coded tags rather than the actual group name. A power analysis was performed prior to the study to estimate sample-size. We hypothesized that HC would reduce IR injury by 50%. Using an SD at ± 30% for the surgery and considering a power at 0.8, we calculated that n = 6 animals/group was necessary to validate a significant effect of the carbo-loading. Animals with pre-existing conditions (malocclusion, injury, abnormal weight) were not operated or excluded from the experiments upon discovery during dissection (kidney disease, tumor etc.). All experiments were analyzed using Prism 9.5.1 (GraphPad Software, USA). Data were presented as mean ± SD and statistical significance was evaluated using Student's t test, one- or two-way ANOVA and multiple comparisons were analyzed using Tukey's and Sidak's post-hoc test. Correlation analyses were determined using Linear Regression Test. A P value inferior or equal to 0.05 was defined as statistically significant. Correlation analysis were determined using Linear Regression Test. A P value inferior or equal to 0.05 was defined as statistically significant. Artworks in figures S1A and S9 were created with BioRender.com (Academic License Terms, www.biorender.com).

**Reporting summary**
Further information on research design is available in the Nature Portfolio Reporting Summary linked to this article.

## Data availability
All custom scripts are available on Figshare or at https://github.com/Longchamp-Lab/agius-et-al-nature-comm-2023. All other relevant data are available from the corresponding author on request. All data from Department of Agriculture National Health and Nutrition Examination Survey are available at https://www.cdc.gov/nchs/nhanes/.

The publicly available sequencing data generated in this study have been deposited in the National Center for Biotechnology Information Gene Expression Omnibus (GEO) and are accessible through the GEO Series accession number GSE39313 and through SRA accession PRJNA851959.

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

## Acknowledgements

We thank our deeply regretted friend and mentor James R. Mitchell. This work was supported by the Swiss National Science Foundation to FL (SNSF 320030_182658) to FA and SD (SNSF-310030_176158), and AL (SNSF PZ00P3-185927), the Mercier Foundation to AL, the Mendez National Institute of Transplantation Foundation to AL, the Novartis foundation to AL and FA, the Leenaards Foundation to AL, the University of Lausanne (projet de recherche interdisciplinaire) to DG and AL, the Fondation pour la recherche en chirurgie thoracique et vasculaire to FA, SD and AL, the Union des Sociétés Suisses des Maladies Vasculaires to SD. We are grateful to the Mouse Pathology Facility (MPF) and the Cellular Imaging Facility (CIF) of the University of Lausanne for their support and expertise.

## Author contributions

T.A., R.E., A.ly, M.R.M., K.K., A.F., M.L., D.L., S.S., D.G., F.L., H.Y., J.F.M., K.U., A.O., S.J.M., F.A., S.D. and A.L. participated in research design. T.A., R.E., A.ly, M.R.M., K.K., A.F., L.S., M.L., D.L., S.S., D.G., F.L., H.Y., J.F.M., K.U., A.O., S.J.M., F.A., S.D. and A.L. participated in the writing of the paper. T.A., R.E., A.ly, M.R.M., K.K., A.F., L.S., M.L., D.L., S.J.M., F.A., S.D. and A.L. participated in the performance of the research. S.S., D.G., F.L., H.Y., J.F.M., K.U., A.O., S.J.M., F.A., S.D. and A.L. contributed new reagents or analytic tools. T.A., R.E., A.ly, M.R.M., K.K., A.F., M.L., D.L., S.J.M., F.A., S.D. and A.L. participated in data analysis. D.G., F.L., F.A., S.D., and A.L. obtained funding.

## Competing interests

The authors declare no competing interests.

## Additional information

[1]Department of Vascular Surgery, University Hospital of Lausanne (CHUV), Lausanne, Switzerland. [2]Transplant Center, Department of Surgery, Massachusetts General Hospital, Harvard Medical School, Boston, MA, USA. [3]Center for Engineering in Medicine, Department of Surgery, Massachusetts General Hospital, Harvard Medical School, Boston, MA, USA. [4]Lewis-Sigler Institute for Integrative Genomics, Princeton University, Princeton, NJ, USA. [5]Laboratory of Nephrology, Department of Internal Medicine Specialties and Department of Cell Physiology and Metabolism, University of Geneva, Geneva, Switzerland. [6]Service of Nephrology, Department of Internal Medicine Specialties, University Hospital of Geneva, Geneva, Switzerland. [7]Transplantation Center, Lausanne University Hospital (CHUV), University of Lausanne (UNIL), Lausanne, Switzerland. [8]Division of Intensive Care, Department of Acute Medicine, University Hospital of Geneva, Geneva, Switzerland. [9]Department of Radiology and Medical Informatics, University of Geneva, Geneva, Switzerland. [10]Center for Biomedical Imaging (CIBM), Geneva, Switzerland. [11]Department of Biomedical Sciences, University of Lausanne, Lausanne, Switzerland. [12]These authors contributed equally: Thomas Agius, Raffaella Emsley. ✉e-mail: alban.longchamp@chuv.ch

