## [Peer Review File · Nature Communications]

Short-Term Hypercaloric Carbohydrate Loading Increases Surgical Stress Resilience by inducing FGF21REVIEWER COMMENTS

Reviewer #1 (Remarks to the Author):

This is an extremely interesting piece of work that has the potential to transform surgical treatment. Previous work by Jay Mitchell has shown that protein depletion prior to surgery can promote better recovery from surgery; however this is not an extremely feasible intervention. Here, the authors show that protein dilution as a consequence of carbohydrate loading can have similar benefits, finding that resilience to surgical stress is improved. Moreover, they show that the mechanism of this effect is mediated by the hormone FGF21. The message here is timely and of high interest, as carbohydrate loading is a translationally easy way to promote surgical stress resilience, and suggests the possibility of future drugs to activate this pathway as well.

Minor edits that could improve the manuscript:

1. Previous work by Mitchell, Lamming, and Simpson suggest that PR has a threshold around 9-10% of dietary protein calories, with PR being induced below this. This is in agreement with the results here that LP diets do not further protect from surgical resilience when combined with carbohydrate loading. The idea of a threshold should be discussed, and Figure 2 expanded to show the % protein intake in both control and HC groups - the existing figure 2D is zoomed too far out to see what the numbers for protein are
2. Interestingly the term protein leverage is not used - this might make it more accessible to readers from many backgrounds.
3. PMID: 35108511 suggests that FGF21 does not regulate protein appetite in B6 females - the authors should discuss evidence here that FGF21 is differently/same regulated across sexes.
4. Aside from Figure 1, main figure legends do not mention sex of mice - should label male or female as appropriate.

Reviewer #2 (Remarks to the Author):

What are the noteworthy results?

the study grounds a common habit, i.e. giving energy rich food before surgery.

Will the work be of significance to the field and related fields? How does it compare to the established literature?

The study is original in the sense that grounds with an interesting pathophysiology approach a common practice. There are however some issues that could be better discussed, and which I listed below.

Does the work support the conclusions and claims, or is additional evidence needed?

yes it is well consistent

Are there any flaws in the data analysis, interpretation and conclusions? Do these prohibit publication or require revision?

some revision may be needed (see below)

Is the methodology sound? Does the work meet the expected standards in your field?

Is there enough detail provided in the methods for the work to be reproduced?

I am a clinician and I analyzed it from the point of view of a clinician interested in metabolism and diet. With this caveat, the work seems well done.

Some suggestions:

1. While giving drinks is often performed and easy, this is not necessarily the only way to have a carbohydrate load. The authors should comment on this.

In some countries (I agree without any reason) the use is to give protein rich nutritional oral supplements (even in patients with chronic kidney disease); while there is no evidence, the authors should consider commenting about this.

2. the fact that carbohydrate intake reduces the intake of other nutrients is quite well known, and one example may be peritoneal dialysis, in which glucose absorption reduces

protein intake. The authors should expand a little this part.

3. Acute carbohydrate loading might be better defined. In this case the carbohydrate loading lasts for several days. Does it really qualify as acute? To me it is short-term, more than acute

4. The shift from mice to men is not fully clear. This is a major point. I suggest either to stick to mice or to better discuss the shift mice to men, also giving an idea of the equivalent "human" doses.

Reviewer #1 (Remarks to the Author):

This is an extremely interesting piece of work that has the potential to transform surgical treatment. Previous work by Jay Mitchell has shown that protein depletion prior to surgery can promote better recovery from surgery; however this is not an extremely feasible intervention. Here, the authors show that protein dilution as a consequence of carbohydrate loading can have similar benefits, finding that resilience to surgical stress is improved. Moreover, they show that the mechanism of this effect is mediated by the hormone FGF21. The message here is timely and of high interest, as carbohydrate loading is a translationally easy way to promote surgical stress resilience, and suggests the possibility of future drugs to activate this pathway as well. We thank the reviewer for their positive comments on our manuscript and suggestions for improvements.

Minor edits that could improve the manuscript:

1. Previous work by Mitchell, Lamming, and Simpson suggest that PR has a threshold around 9-10% of dietary protein calories, with PR being induced below this. This is in agreement with the results here that LP diets do not further protect from surgical resilience when combined with carbohydrate loading. The idea of a threshold should be discussed, and Figure 2 expanded to show the % protein intake in both control and HC groups - the existing figure 2D is zoomed too far out to see what the numbers for protein are

Previous work from the above-mentioned team demonstrated that PR with less than 10% of dietary protein from calories produced the strongest response (body composition, Insulin PMID: 35977507), noting some signaling pathways (e.g ATF4, FGF21) seem only to be triggered when protein is less than 8% (PMID: 30385734 and 35977507). While we haven't rigorously tested the effect of various levels of PR, we agree with the reviewer that there is a threshold effect, rather than a linear effect of PR in the the context of protection from surgical stress. Our data and data from other labs suggest that most of the benefits occur with protein less than 8-10 % with only modest improvements after that. As depicted in the figure below, the protection from surgical stress was mostly consistent at 6.4% protein energy intake and below. Figure 2D was modified to show the % protein intake on the bar graph.

The idea of a threshold is discussed in the revised manuscript on age page 16-17. "The response to dietary protein is also not linear. Previous work demonstrated that protein restriction has a threshold around 8-10% of dietary protein calories to produce the strongest improvements in body composition, insulin sensitivity⁴⁰ and induction of FGF21²³. In our study, when comparing carbohydrate loading (3.4% dietary protein calories) versus a low protein diet (6.4% dietary protein calories), the protection from surgical stress was similar. This is consistent with the idea of a threshold and non-linear effects^{40, 50}. However, a study comparing a range of sucrose concentrations during carbohydrate loading would be of interest to better investigate the potential for dose response to carbohydrate loading.

Figure Reviewing 1. Heatmap of the response to protein dilution at the indicated regimen for one week.

2. Interestingly the term protein leverage is not used - this might make it more accessible to readers from many backgrounds.

We thank the reviewer for pointing this out. Indeed, protein intake is strongly regulated by FGF21, and we do observe increased intake in animals fed a low protein diet with normal water. However, our data do not strongly support the protein leverage concept because we do not observe protein leverage in mice fed low protein with sucrose water. We believe the ability of sucrose water to induce protein dilution while overriding protein leverage is extremely interesting. We do think this is an important area for follow up and now discuss protein leverage in the context of our findings (page 16).

“Rodents have a dominant protein appetite, which drives food intake to reach a protein target 50 via an FGF21-dependent mechanism 36, 39. Thus, a protein-restricted diet typically results in the overconsumption of total energy. This response is called protein leverage 50. A limited body of evidence also supports the protein leverage hypothesis in humans, at least down to 10% of protein energy 51. While we do replicate the protein leverage hypothesis in mice fed low protein with normal water, we interestingly observe that mice given sucrose water do not increase food intake on a low protein diet. This suggests the intriguing possibility that liquid carbohydrate intake may override protein leverage. Future studies may test the resulting hypothesis that FGF21 action on drinking behavior may override its action on feeding behavior.”

3. PMID: 35108511 suggests that FGF21 does not regulate protein appetite in B6 females - the authors should discuss evidence here that FGF21 is differently/same regulated across sexes.

This is in fact very interesting, in accordance with the Lamming group (35108511) we observed that while FGF21 was required for the protection from surgical stress, it is dispensable for the regulation of water and food intake by liquid carbohydrate (Figure 4). On the other hand, exogenous FGF21 administration was sufficient to promote food intake. Additionally, we found that 1 week of carbohydrate loading significantly increased blood FGF21 (see revised Extended Data Fig. 5b and c) in both female and aged mice. While this is consistent with our hypothesis

that FGF21 mediates carbohydrate loading induced protection from surgical stress, this somehow differs from previous findings in which the correlation between FGF21 and protein intake was sex and strain dependent (PMID: 35108511). Consistently with differences between the response to low protein diets and carbohydrate loading, mice lacking FGF21 fail to exhibit metabolic response to protein restriction (PMID 35393401). Two possibilities may explain our observation that liquid carbohydrate increases FGF21 in females while low protein diets do not; first, we achieve a greater protein restriction with liquid carbohydrate and females require a greater restriction for induction or second, that mechanisms other than just protein restriction may regulate FGF21 upon liquid carbohydrate intake. The discussion was revised accordingly page 15 “To further support the generality of our findings, the response to carbohydrate loading including protection from surgical stress, and circulating FGF21 was consistent across sex and in older mice. Interestingly, this somehow differs from previous work in which the correlation of FGF21 with protein intake was sex and strain-dependent⁴⁹. Our carbohydrate loading protocol does induce greater protein restriction than diets used previously (~3.5% vs 6% protein intake), so it is possible that females require greater protein restriction to induce FGF21.”

4. Aside from Figure 1, main figure legends do not mention sex of mice - should label male or female as appropriate.

Age and sex are now described clearly in the figure legends

Reviewer #2 (Remarks to the Author):

What are the noteworthy results?

the study grounds a common habit, i.e. giving energy rich food before surgery.

Will the work be of significance to the field and related fields? How does it compare to the established literature?

The study is original in the sense that grounds with an interesting pathophysiology approach a common practice. There are however some issues that could be better discussed, and which I listed below.

Does the work support the conclusions and claims, or is additional evidence needed?

yes it is well consistent

Are there any flaws in the data analysis, interpretation and conclusions? Do these prohibit publication or require revision?

some revision may be needed (see below)

Is the methodology sound? Does the work meet the expected standards in your field?

Is there enough detail provided in the methods for the work to be reproduced?

I am a clinician and I analyzed it from the point of view of a clinician interested in metabolism and diet. With this caveat, the work seems well done.

We thank the reviewer for their positive comments on our manuscript and constructive feedback.

Some suggestions:

1. While giving drinks is often performed and easy, this is not necessarily the only way to have a carbohydrate load. The authors should comment on this.

This is true: in addition to drinks, carbohydrates can be acutely administered i.v. or during peritoneal dialysis as discussed below. In our study, the rationale for using carbohydrate drinks also came from current enhanced-recovery ERAS®)protocols , which encourage the consumption of carbohydrate loaded fluids before surgery.

In some countries (I agree without any reason) the use is to give protein rich nutritional oral supplements (even in patients with chronic kidney disease); while there is no evidence, the authors should consider commenting about this.

We thank the reviewer for pointing this out and we fully agree. Despite the evidence, there is still some institutions administering high protein drinks, to “support the healing process” or the “immune system” PMID: 30817349). We have included these points in the introduction on page 3 : “Moreover, within ERAS®, carbohydrate loading benefits are confounded by numerous interventions (e.g. analgesia, early removal of catheters, thromboprophylaxis, etc. ⁷), and ERAS® presurgery drinks often contain high amounts of protein (up to 80g/L) making it challenging to evaluate its efficacy.”

2. the fact that carbohydrate intake reduces the intake of other nutrients is quite well known, and one example may be peritoneal dialysis, in which glucose absorption reduces protein intake. The authors should expand a little this part.

Dialysis solution comes in 1.5 percent, 2.5 percent, and 4.25 percent dextrose concentrations, while a 50% sucrose (glucose and fructose) was used in our study. Additionally, our carbohydrate loading was only performed for 1 week (versus months to years of peritoneal dialysis). While peritoneal dialysis is well known to induce hyperglycemia in some patients, we are not aware of data suggesting that it might reduce food intake directly. On the other hand, our preliminary data indicate that energy metabolism, insulin sensitivity and glucose tolerance are all improved after 1-week carbohydrate loading. This is consistent with a time dependent effect, where short term highly concentrated carbohydrate administration is beneficial but might be detrimental if given for a prolonged period of time.

The interaction between protein and carbohydrate is very interesting indeed. Humans and rodents have a dominant protein appetite, which disproportionately drives food intake (the so-called protein leverage). Hence, a low protein carbohydrate-rich diet increased food intake and body weight but also generated better cardiometabolic outcomes (lifespan, lipoproteins, glycaemic status). On the contrary, in westernized nations, substantial proportions of calories are derived from readily digestible, palatable carbohydrates, which dilute protein in the food supply and encourage increased calorie intake, promoting obesity and metabolic disorders. Additionally, the source of carbohydrate seems to be of crucial importance. Low protein, high carbohydrate diets promote the healthiest metabolic outcomes when carbohydrate comprises resistant starch, yet the worst outcomes were with a 50:50 mixture of monosaccharides fructose and glucose (PMID: 34099926)

3. Acute carbohydrate loading might be better defined. In this case the carbohydrate loading lasts for several days. Does it really qualify as acute? To me it is short-term, more than acute. The term 'acute' was changed to "one week" when referring to carbohydrate loading for more clarity.

4. The shift from mice to men is not fully clear. This is a major point. I suggest either to stick to mice or to better discuss the shift mice to men, also giving an idea of the equivalent "human" doses.

The rationale for using a large survey dataset and analyzing human feeding behaviors was improved in the revised manuscript (page 8).

REVIEWERS' COMMENTS

Reviewer #1 (Remarks to the Author):

The authors have made appropriate revisions and should be applauded for this insightful work, which is both timely and will be of interest to readers from a broad spectrum of biology.

Reviewer #2 (Remarks to the Author):

I read with interest the revised version of the paper.

I would appreciate however some higher precisions.

This was the first question

1. While giving drinks is often performed and easy, this is not necessarily the only way to have a carbohydrate load. The authors should comment on this.

The answer is quite a tautology:

This is true: in addition to drinks, peritoneal dialysis as discussed below. In our study, the rationale carbohydrates can be acutely administered i.v. or during for using carbohydrate drinks also came from current enhanced-recovery ERAS®) protocols , which encourage the consumption of carbohydrate loaded fluids before surgery.

My further comments:

acute administration of carbohydrates iv is not clinically sound, or feasible in the clinics (and should in any case be defined).

It is not at all clear how peritoneal dialysis could be used for this mean! high sugar content may lead to an unwanted fluid loss, and to other side effects.

The authors did not talk of oral supplies in different forms.

I did not find any comment in the paper about this.

The authors should clarify this in the text.

This was the second question:

2. In some countries (I agree without any reason) the use is to give protein rich nutritional oral supplements (even in patients with chronic kidney disease); while there is no evidence, the authors should consider commenting about this.

We thank the reviewer for pointing this out and we fully agree. Despite the evidence, there is still some institutions administering high protein drinks, to “
or the support the healing process” “immune system” PMID: 30817349). We have included these points in the introduction on page 3:“

The comment only partly answers to the question:

Moreover, within ERAS[®], carbohydrate loading benefits are confounded by numerous interventions (e.g. analgesia, early removal of catheters, thromboprophylaxis, etc. 7), and ERAS[®] presurgery drinks often contain high amounts of protein (up to 80g/L) making it challenging to evaluate its efficacy.”

I would appreciate a comment on the lack of evidence on the advantages of protein rich food, not on the fact that they may be confounders.

3. Acute carbohydrate loading might be better defined. In this case the carbohydrate loading lasts for several days. Does it really qualify as acute? To me it is short-term, more than acute.

The answer, "The term 'acute' was changed to “one week” when referring to carbohydrate loading for more clarity." is partly satisfactory.

Why one week? this is not "acute in humans"; the term acute is still frequently used in the paper. A clear definition would be useful.

Previous comment:

4. The shift from mice to men is not fully clear. This is a major point. I suggest either to stick to mice or to better discuss the shift mice to men, also giving an idea of the equivalent "human" doses.

This is what the authors added to the text: (page 8)

" Total carbohydrate and protein intake was analyzed from the National Health and Nutrition Examination Survey (NHANES) from 2005 to 2012 including two 24-h dietary recalls per person on a representative sample of the United States population every 2 years (four distinct cycles of data collection). Using this large dataset, we..."

The answer is unsatisfactory; I strongly suggest the authors limit their discussion to mice, and use the available literature evidence on humans in the discussion.

Reviewer #1 (Remarks to the Author):

The authors have made appropriate revisions and should be applauded for this insightful work, which is both timely and will be of interest to readers from a broad spectrum of biology.

Thank you

Reviewer #2 (Remarks to the Author):

1. acute administration of carbohydrates iv is not clinically sound, or feasible in the clinics (and should in any case be defined).

It is not at all clear how peritoneal dialysis could be used for this mean! high sugar content may lead to an unwanted fluid loss, and to other side effects.

The authors did not talk of oral supplies in different forms.

I did not find any comment in the paper about this.

The authors should clarify this in the text.

Alternative approach is now discussed, see page 16-17 "Here, carbohydrate loading consisted of an *ad libitum* solution of 50% sucrose in water for one week. This slightly differs from ERAS® protocols, classically recommending 100 g of carbohydrate the evening before surgery, and 50 g of carbohydrate until 2 hours before surgery. Alternatively, healthy individuals have undergone 3-7 days of high carbohydrate loading diet (approximately 70% of calories from carbohydrate), at the expense of protein. The latter is similar to most murine dietary protein restriction regimen, that replaced dietary protein with carbohydrates utilized complex carbohydrates derived from starch"

2. I would appreciate a comment on the lack of evidence on the advantages of protein rich food, not on the fact that they may be confounders.

The introduction was modified as requested page 4 "Moreover, ERAS® pre-surgery drinks often contain high amounts of protein (up to 80g/L). Interestingly, the benefit of high protein diet before surgery has not been demonstrated. Inversely, high animal protein intake was positively associated with cardiovascular mortality⁷."

3. "The term 'acute' was changed to "one week" when referring to carbohydrate loading for more clarity." is partly satisfactory.

Why one week? this is not "acute in humans"; the term acute is still frequently used in the paper. A clear definition would be useful.

To improve the clarity of the paper, the term "acute" was removed from the manuscript. The rationale for the "one-week" regimen is indeed a very interesting and is discussed page 18 "The duration of carbohydrate loading required for the onset of stress resistance benefits is unknown. Here, we show that only one week of carbohydrate loading is-sufficient for the observed benefits in mice with maximal FGF21 induction already achieved after two days, indicating that this relatively short duration of carbohydrate loading might be sufficient to yield the observed benefits. Whether or not longer periods of carbohydrate loading will further increase protection remains to be experimentally determined."

4. The shift from mice to men is not fully clear. This is a major point. I suggest either to stick to mice or to better discuss the shift mice to men, also giving an idea of the equivalent "human" doses. I strongly suggest the authors limit their discussion to mice, and use the available literature evidence on humans in the discussion.

We thank the reviewer for emphasizing the limitation of the human data. The discussion was significantly revised to highlight the limitations of the presented data. The term "human" was minimized in the revised version. We further acknowledge now clearly the limitations of our

human data, emphasizing the absence of causality and the descriptive nature of the findings. See page 16.

“The observational human data from the USDA NHANES suggested an inverse association between sucrose and protein intake, indicating the potential for conserved appetitive mechanisms. However, the metabolic response and the extent to which carbohydrate loading may result in favorable outcomes in humans are still unknown. As with any epidemiologic study, the association of macronutrient intake can be confounded by any other variable (lifestyle, other dietary factors, comorbidities, medications, etc.). Although protein energy intake was treated as a single variable, it truly is a combination of amino acid ratios that cannot be fixed, even in mice which limit drawing conclusions about the effects of dietary protein in humans.”

Despite the above-mentioned limitations, we strongly believe that the presented human data is crucial to the manuscript, considering the potential translational significance of the findings.